# Learning to Watch: Active Video Anomaly Understanding via Interleaved Policy Optimization

**Mengjingcheng Mo** [1] **Jiaxu Leng** [1 2] **Xinbo Gao** [1]

## Abstract

Video anomaly understanding (VAU) relies on sparse, context-dependent cues. However, existing passive paradigms suffer from observational aliasing, where static sampling fails to disambiguate semantically distinct events. To overcome this, we propose $Anom\text{-}\pi$, a closed-loop framework that reconceptualizes video understanding as an active sequential decision-making process within a dynamic environment. Inspired by human video-reviewing behavior, this framework unifies internal cognitive reasoning and strategic evidence acquisition into an interleaved policy, utilizing temporal atomic operators such as local backtracking, temporal expansion, and fine-grained sampling to endow the model with perceptual proactivity. To learn such complex interaction strategies under video-level weak supervision, we design Interactive Direct Preference Optimization (iDPO) to achieve trajectory-level policy alignment, guided by an Active Evidence Inquiry (AEI) utility that balances task success, informative evidence acquisition, and interaction cost. This approach enables the agent to learn to actively disambiguate hypotheses while suppressing redundant exploration. Extensive experiments demonstrate that our framework, with only 2B parameters, achieves highly competitive performance, significantly outperforming state-of-the-art large-scale VAU models in complex scenarios.

## 1. Introduction

Video anomaly understanding (VAU) aims to build a general anomaly recognition capability for open-world diver-

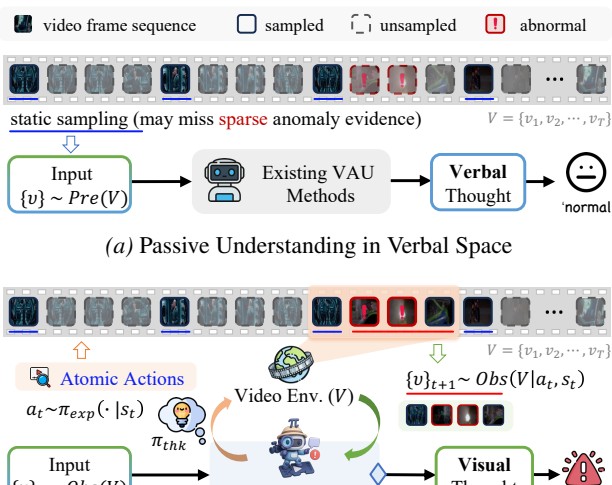

*(a)* Passive Understanding in Verbal Space

*(b)* Active Exploration in Visual Space

*Figure 1.* **Comparison of passive and active evidence acquisition for video anomaly understanding.** (a) Existing VAU methods use static sampling in the verbal space, which may miss sparse anomaly evidence. (b) Anom-$\pi$ actively explores the video in the visual space to gather informative observations and detect anomalies.

sity. The task requires localizing anomalous events in complex video streams and producing verifiable evidence. Discriminative cues for anomalies are typically sparse, making ambiguous cases rely on temporal context for disambiguation. However, incorporating broader context introduces substantial redundant background that can drown out the few critical cues (Chen et al., 2025). General-purpose large language models (LLMs) may then fail to reliably aggregate evidence into verifiable decision chains, yielding suboptimal decisions. This highlights a bottleneck beyond reasoning, namely how evidence is selected and coupled with the reasoning process.

This bottleneck is particularly acute under weak supervision (Wu et al., 2020; Sultani et al., 2018), a practical regime where training provides only video-level labels while evaluation requires frame-level detection. In this setting, a key challenge is how to shape LLM-based anomaly decision behavior using only coarse supervision. Existing LLM-

---

[1]School of Computer Science and Technology, Chongqing University of Posts and Telecommunications, Chongqing, China [2]Chongqing College of Artificial Intelligence, Chongqing, China. Correspondence to: Xinbo Gao <gaoxb@cqupt.edu.cn>.

*Proceedings of the 43rd International Conference on Machine Learning*, Seoul, South Korea. PMLR 306, 2026. Copyright 2026 by the author(s).

based methods can be broadly categorized into three routes. Training-free approaches (Zanella et al., 2024; Li et al., 2026a; Yang et al., 2026) rely on prompt-based reasoning without data-driven optimization, using frozen LLM pipelines to score anomalies. In contrast, supervision-intensive approaches (Du et al., 2024; Zhang et al., 2025; Huang et al., 2026) construct anomaly-focused instruction data and fine-tune large multimodal models for detection and explanation. Alternatively, some methods (Tang et al., 2024; Huang et al., 2025) train specialized anomaly detectors for detection and use LLMs mainly for interpretation or caption generation.

These methods remain passive and open-loop, with evidence fixed before inference, as illustrated in Fig. 1a. Even with multi-step reasoning, the pipeline is one-way from perception to cognition, without feedback to refine evidence acquisition under uncertainty. One approach (Zhang et al., 2025; Chen et al., 2025) is to improve evidence selection with auxiliary scorers or lightweight models that estimate importance or anomaly likelihood. Yet evidence acquisition is still decoupled from the final decision, and these proxy scores can break under domain shift. More fundamentally, the open-loop setting prevents learning an understanding-driven evidence-accumulation strategy, leaving the model unable to acquire more evidence under uncertainty or terminate rationally once sufficient, which reflects limited perceptual autonomy.

To address this mismatch, we propose Anom-$\pi$, which reframes anomaly understanding from static classification into an active, closed-loop hypothesis-verification process (Fig. 1b). We cast anomaly understanding as sequential decision-making and treat THINK, INTERACT, and FINAL as peer discrete actions, yielding an interleaved policy $\pi$. Specifically, INTERACT allows the agent to actively gather evidence through operations such as local backtracking, temporal expansion, and fine-grained sampling. Driven by the gathered information, FINAL then serves as an evidence-sufficiency-based termination action rather than a post-hoc rule driven by external thresholds. The policy can invoke reproducible atomic observation operators to probe ambiguity and terminate once evidence is sufficient, thereby closing the perception loop and reducing high-confidence errors induced by open-loop pipelines.

To learn $\pi$ without fine-grained action or state supervision, we adopt preference-based policy learning. For each video, we sample multiple trajectories and construct chosen/rejected preference pairs under video-level weak-label constraints, turning interaction choices and termination timing into scalable training signals. We define preferences by the value of information, favoring interactions that increase hypothesis separability while suppressing ineffective or redundant observations. Concretely, we instantiate this

principle with an Active Evidence Inquiry (AEI) utility that combines task success, intrinsic inquiry incentives, and interaction cost. We then optimize an Interactive Direct Preference Optimization (iDPO) objective to achieve trajectory-level alignment, bypassing explicit reward modeling and the inherent instabilities of on-policy policy gradients. Importantly, rejected trajectories provide a contrastive signal that suppresses near-miss behaviors, where outcomes appear correct but evidence acquisition or stopping decisions are flawed, improving cross-scenario generalization.

Our main contributions are as follows. (1) We propose Anom-$\pi$, which reformulates video anomaly understanding as a closed-loop sequential decision-making process with active interaction. (2) We design a set of reproducible and composable atomic observation operators, and implement a controllable interact interface via tool calls. (3) We propose Interactive Direct Preference Optimization (iDPO), guided by an Active Evidence Inquiry (AEI) utility, to align the interleaved policy $\pi$ at the trajectory level. (4) We validate the effectiveness and generalization of the learned interleaved policy on multiple benchmarks, and show that active evidence acquisition improves evidence discovery and decision reliability.

## 2. Related Work

**Video Anomaly Detection.** The task is challenging due to sparse anomalies, costly temporal annotations, and open-set semantics (Sultani et al., 2018; Wu et al., 2020; Acsintoae et al., 2022). Most practical pipelines are weakly supervised and learn clip-level anomaly scores from video-level labels under the multi-instance learning paradigm (Tian et al., 2021), using uncertainty regulation (Zhou et al., 2023), self-distillation (Ristea et al., 2024), or multimodal aggregation (Majhi et al., 2025). Recent work further incorporates semantic priors by leveraging clip-level semantics (Wu et al., 2024; Chen et al., 2024a; Wang & Chen, 2025) or vision-language representations (Huang et al., 2025; Li et al., 2025b; 2023; 2025a), improving discriminability and interpretability under coarse supervision. Overall, existing methods typically follow a fixed observation protocol with predefined sampling and single-pass inference, which may degrade when anomalies are sparse or evidence is highly contextual.

**Video Anomaly Understanding.** Recent studies move beyond scoring and localization toward video anomaly understanding, producing anomaly categories (Huang et al., 2025) and natural language rationales via vision-language and language models (Zanella et al., 2024). Training-free paradigms (Shao et al., 2025; Li et al., 2026a; Yang et al., 2026) can reason over temporal cues with frozen models (Liu et al., 2024; Chen et al., 2024b), but their reliability is limited by expert-guided orchestration

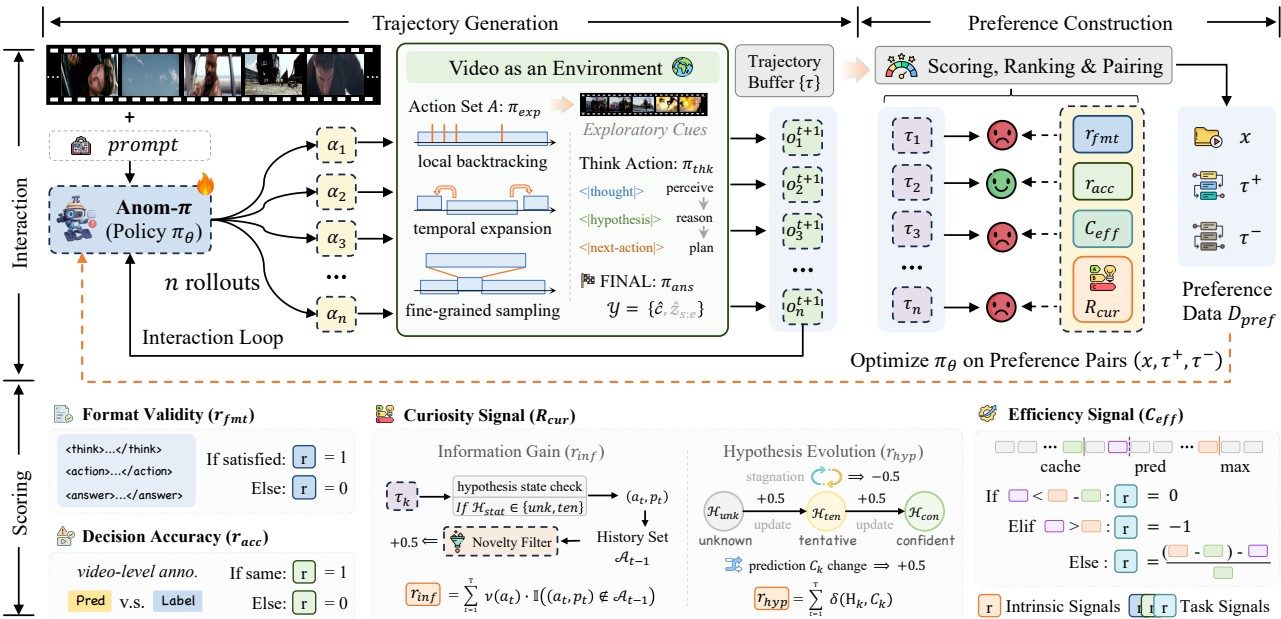

*Figure 2.* **Overview of Anom-$\pi$.** **Left:** closed-loop active inquiry that alternates deliberation (THINK) with evidence acquisition (BACKTRACK/EXPAND/SAMPLE) before terminating with FINAL to output clip-level hypotheses. **Right:** trajectory-level preference construction and optimization via iDPO from ranked rollouts.

pipelines. Complementary approaches improve sensitivity to sparse anomalies through lightweight supervision, including prompt learning (Yang et al., 2024; Ye et al., 2025), lightweight adapters (Zhang et al., 2025), and token-level selection (Tang et al., 2024) or alignment (Chen et al., 2025) that steers model attention toward anomaly evidence. Recent works also explore reinforcement-style reasoning optimization for VAU and related anomaly-understanding tasks (Huang et al., 2026; Zhu et al., 2025; Mo et al., 2026; Yu et al., 2026; Li et al., 2026b; Kang et al., 2026). Orthogonal to these reasoning-optimization efforts, we focus on making evidence acquisition itself learnable by reformulating VAU as an interaction policy that dynamically adapts its observation strategy based on evolving anomaly hypotheses, synergistically coupling active perception with iterative reasoning.

## 3. Method

### 3.1. Problem Formulation

We view video anomaly understanding as *active inquiry* in a dynamic video environment. Given an untrimmed video $V = \{x_t\}_{t=1}^{T}$, we form overlapping clips with a sliding window of length $L$ and stride $\Delta$: $c_i = \{x_t\}_{t=t_i}^{t_i+L-1}$, where $t_i = 1 + (i-1)\Delta$. Each clip $c_i$ defines an episode where the agent navigates an information space via a unified action space $\mathcal{A} = \mathcal{A}_{\text{inq}} \cup \{\text{THINK}, \text{FINAL}\}$, with $\mathcal{A}_{\text{inq}} = \{\text{BACKTRACK}, \text{EXPAND}, \text{SAMPLE}\}$. At step $n$, the agent conditions on the trajectory $\tau_n = (o_1, a_1, e_1, \ldots, o_n)$,

where $o_n$ encapsulates the current belief (including a structured hypothesis state $\text{hyp}_n$) and $e_n$ is the evidence returned by investigative actions. The episode terminates when $a_n = \text{FINAL}$, yielding up to $M$ hypotheses $(\hat{y}_{i,m}, \hat{b}_{i,m}, \hat{p}_{i,m})$, where $\hat{y}_{i,m} \in \mathcal{Y}$ is the category, $\hat{b}_{i,m} = (\hat{t}_{i,m}^{\text{st}}, \hat{t}_{i,m}^{\text{ed}})$ is the temporal range (within the clip), and $\hat{p}_{i,m} \in [0,1]$ is the confidence. We define the clip-level anomaly confidence as $\hat{z}_i = \max_m \hat{p}_{i,m}$. For frame-level scoring, each clip episode produces a local score vector over its $L$ frames. If the episode predicts normal or has no valid anomalous range, all local scores are 0; otherwise, frames inside the predicted anomalous range(s) receive score 1, and the remaining frames in the same clip receive score 0.5. The final score of a video frame is obtained by averaging the scores from all overlapping clips covering that frame.

### 3.2. Closed-Loop Interactive Inference

Unlike one-shot, open-loop mappings, Anom-$\pi$ performs *closed-loop* inference by alternating epistemic *deliberation* and *evidence queries* (Fig. 2, left). This mechanism enables hypothesis revision conditioned on newly acquired evidence and allows the agent to defer decisions when information is insufficient.

At each step $n$, the agent selects an action $a_n \in \mathcal{A}$. Investigative actions $\mathcal{A}_{\text{inq}}$ acquire new evidence $e_n$ by changing the observation view. These actions are designed to mimic human video-reviewing behaviors (*learning to watch*), rather than being arbitrary tool calls: BACKTRACK focuses on

a subset of frames within the current clip to zoom in on potentially informative cues; EXPAND queries extended temporal context beyond the current clip to disambiguate; SAMPLE densely probes around suspicious moments for higher-resolution cues. The epistemic action THINK does not interact with the environment; it updates the structured hypothesis state $\mathrm{hyp}_n$ by integrating newly acquired evidence. The termination action FINAL ends the episode and outputs up to $M$ hypotheses $(\hat{y}_{i,m}, \hat{b}_{i,m}, \hat{p}_{i,m})$ with a one-sentence explanation.

Closed-loop interaction follows the iterative pattern THINK $\rightarrow a_{\mathrm{inq}} \in \mathcal{A}_{\mathrm{inq}} \rightarrow$ THINK. In each round, the current hypothesis $\mathrm{hyp}_n$ guides which information is missing, an investigative action is chosen accordingly, and the returned evidence $e_n$ is used to recalibrate the hypothesis. The stopping decision $a_n =$ FINAL is part of the learned policy $\pi_\theta$ rather than an external threshold. In practice, the agent tends to stop when its hypothesis becomes confident and additional queries yield diminishing information value. We provide a value-of-information (VoI) analysis in Section 4.

### 3.3. Incentivizing Active Evidence Inquiry

To guide the policy $\pi_\theta$ to navigate efficiently in the unified action space, we design an Active Evidence Inquiry (AEI) utility that combines task-driven rewards, intrinsic inquiry incentives, and interaction costs. For a trajectory $\tau$, we define

$$U(\tau) = R_{\mathrm{task}}(\tau) + \lambda \, R_{\mathrm{aei}}(\tau) - \eta \, C(\tau), \qquad (1)$$

where $R_{\mathrm{task}}$ encourages task success, $R_{\mathrm{aei}}$ incentivizes informative evidence acquisition, $C(\tau)$ penalizes excessive interaction, and $\lambda$ and $\eta$ are balancing coefficients.

**Task-driven rewards ($R_{\mathbf{task}}$).** We use external rewards to ensure the final decision is both accurate and well-formed. In practice, $R_{\mathrm{task}}$ is a weighted combination of two binary signals: *decision accuracy* ($r_{\mathrm{acc}}$; matching clip-level supervision) and *format validity* ($r_{\mathrm{fmt}}$; conforming to the prescribed structured output template).

**Active inquiry incentives ($R_{\mathbf{aei}}$).** To prevent premature guessing under insufficient evidence, we introduce an epistemic drive with two components. We define the intrinsic acquisition return as

$$R_{\mathrm{aei}}(\tau) = \sum_{t=1}^{N} r_{\mathrm{cur}}(\tau_t), \; r_{\mathrm{cur}}(\tau_t) = r_{\mathrm{inf}}(\tau_t) + r_{\mathrm{hyp}}(\tau_t), \; (2)$$

where $t$ indexes interaction steps (actions) and $N$ is the trajectory length. (i) *Information gain* ($r_{\mathrm{inf}}$): following the novelty filter in Fig. 2, let $(a_t, p_t)$ denote the step-$t$ investigative action and its queried location (e.g., temporal

position), and let $\mathcal{A}_{t-1}$ be the set of previously queried action-location pairs. We define

$$r_{\mathrm{inf}}(\tau_t) = v(a_t) \, \mathbb{I}\big[(a_t, p_t) \notin \mathcal{A}_{t-1}\big], \qquad (3)$$

where $v(a_t)$ optionally weights actions by their informativeness. (ii) *Belief refinement* ($r_{\mathrm{hyp}}$): we reward meaningful hypothesis evolution in the structured belief state $\mathrm{hyp}_t$. Concretely, we grant a bonus when (i) the hypothesis state advances (e.g., *unknown→tentative→confident*) or (ii) newly acquired evidence triggers a revision of the predicted category. A simple instantiation is

$$r_{\mathrm{hyp}}(\tau_t) = \alpha \, \mathbb{I}[\mathcal{H}_t \succ \mathcal{H}_{t-1}] + \gamma \, \mathbb{I}[\hat{y}_t \neq \hat{y}_{t-1}], \quad (4)$$

where $\mathcal{H}_t$ denotes the discrete hypothesis status at step $t$ and $\succ$ is an ordering from less to more confident. In our implementation, we assign a reward bonus of $0.5$ for successful refinement.

**Efficiency and rational termination ($C(\tau)$).** We assign interaction costs to reflect the overall computational budget, including the dominant cost of model deliberation. To encourage exploration (i.e., to let the agent query evidence before being penalized) while still accounting for compute, we use an accumulated step-cost with a free "buffer" of $N_{\mathrm{buf}}$ steps:

$$C(\tau) = \sum_{t=1}^{N} c_t, \; c_t = \mathbb{I}\big[t > N_{\mathrm{buf}}\big], \qquad (5)$$

where $N$ is the number of interaction steps and $N_{\mathrm{buf}}$ specifies how many initial steps are not penalized. This removes the explicit weighting hyperparameters (e.g., $c_0, \kappa$) while still keeping a small exploration buffer. As characterized in Theorem 4.4, the policy learns to execute FINAL when the expected value of information (VoI) from any further interaction falls below its corresponding cost, yielding rational stopping without an explicit heuristic threshold.

### 3.4. Policy Optimization via Interactive DPO (iDPO)

Conventional supervised fine-tuning (SFT) maximizes likelihood on a single "best" trajectory, which can be brittle in a large interactive space and may lead to overconfident, poorly calibrated stopping decisions. To learn robust decision-making under varying evidence quality, we adopt *interactive* Direct Preference Optimization (iDPO), building on Direct Preference Optimization (DPO) (Rafailov et al., 2023), which injects contrastive signals directly in the space of closed-loop trajectories.

For each clip episode $c_i$, we perform $n$ independent rollouts under the current policy $\pi_\theta$ to obtain a set of candidate interaction trajectories $\{\tau_1, \ldots, \tau_n\}$. Due to stochastic sampling, these trajectories differ in their evidence-query paths,

deliberation depth, and termination time. We score each trajectory with the AEI utility $U(\tau)$ (Eq. (1)), which integrates task success (e.g., decision accuracy $r_{\text{acc}}$ and format validity $r_{\text{fmt}}$), intrinsic inquiry benefits (e.g., information gain $r_{\text{inf}}$ and belief refinement $r_{\text{hyp}}$), and interaction efficiency via the cost term $C(\tau)$. We then rank the rollouts by $U(\tau)$ and extract preference pairs $(\tau^+, \tau^-)$ such that $U(\tau^+) > U(\tau^-)$. Let $\pi_0$ denote a fixed reference policy (e.g., the SFT initialization). Rather than training an explicit reward model, iDPO directly optimizes the trajectory likelihood ratio via the DPO loss

$$\mathcal{L}_{\text{iDPO}}(\theta) = -\mathbb{E}_{(\tau^+, \tau^-)}\Big[\log \sigma\big(\beta\,(s_\theta(\tau^+) - s_\theta(\tau^-))\big)\Big], \tag{6}$$

where $s_\theta(\tau) = \log \frac{\pi_\theta(\tau)}{\pi_0(\tau)}$, $\sigma(\cdot)$ is the logistic sigmoid, and $\beta$ controls the preference sharpness. For an interaction trajectory $\tau$, the policy likelihood decomposes as $\pi_\theta(\tau) = \prod_{t=1}^{N} \pi_\theta(a_t \mid \tau_t)$, so minimizing Eq. (6) directly updates the per-step query/stop decisions.

The resulting push-pull signals reinforce trajectories that acquire discriminative evidence with fewer queries and terminate rationally, while suppressing redundant queries and premature stopping. The ratio to $\pi_0$ acts as an implicit KL regularizer, anchoring updates and mitigating distributional shift from the reference policy. By contrasting $\tau^+$ and $\tau^-$, iDPO increases the separability between preferred and dispreferred interaction trajectories, echoing the theoretical results in Section 4.

## 4. Theoretical Analysis

This section provides theoretical justification for closed-loop video anomaly understanding: (i) open-loop observation admits an irreducible ambiguity; (ii) interaction can accumulate discriminability across rounds; and (iii) termination follows a value-of-information (VoI) tradeoff. Additional discussion and derivations are deferred to Appendix A.

**Setup.** Let the latent explanation be $\theta \in \Theta$ and the video process be $X \sim p_\theta$. At round $t$, the agent selects an action $a_t \in \mathcal{A}$ and receives an observation $o_t$, updating the history as $h_t = h_{t-1} \oplus (a_t, o_t)$. The termination action $\text{FINAL}(\hat{y})$ ends the episode.

Open-loop methods fix an observation operator $g$ and perform inference only from $Y = g(X)$. We denote the induced distribution of $Y$ under $X \sim p_\theta$ by $p_\theta^g$.

**Theorem 4.1** (Open-loop Barrier). *If there exist $\theta \neq \theta'$ such that $D_{\text{TV}}\big(p_\theta^g, p_{\theta'}^g\big) \leq \varepsilon$, then for any estimator $\hat{\theta}(Y)$ depending only on $Y$, the worst-case error obeys $\inf_{\hat{\theta}} \sup_{\vartheta \in \{\theta, \theta'\}} \Pr_\vartheta\big(\hat{\theta}(Y) \neq \vartheta\big) \geq \frac{1}{2}(1 - \varepsilon)$.*

*Proof sketch.* By Le Cam's two-point method. This theorem implies an irreducible error whenever discriminative cues are not present in the fixed view $Y$.

In closed-loop interaction, future actions depend on intermediate history. Let $\tau = (a_{1:H}, o_{1:H})$ be the interaction trajectory under policy $\pi$, with induced distribution $P_\theta^\pi$. Here $H$ denotes the number of interaction rounds, and $o_t$ represents the returned evidence/observation at round $t$.

**Assumption 4.2** (Per-round Discriminability Gain). Let $\Delta_t(h_{t-1}, a) = D_{\text{KL}}\big(p_\theta(\cdot \mid a, h_{t-1}) \,\|\, p_{\theta'}(\cdot \mid a, h_{t-1})\big)$. There exist a policy $\pi$ and $c > 0$ such that $\mathbb{E}_{h_{t-1}, a_t \sim P_\theta^\pi}[\Delta_t(h_{t-1}, a_t)] \geq c$ for each round $t$. This is an existential assumption: the gain is maintained by an informative policy that selects high-information actions in ambiguous histories, rather than by arbitrary interaction.

Intuitively, if a policy assigns at least probability $\rho$ to informative actions with conditional KL gap at least $\delta$, the expected per-round gain is at least $\rho\delta$; AEI/iDPO serve as empirical proxies that encourage such action selection, rather than a formal guarantee (Appendix A.3).

**Theorem 4.3** (Exponential Error Decay). *Under Assumption 4.2, $D_{\text{KL}}\big(P_\theta^\pi \,\|\, P_{\theta'}^\pi\big) \geq cH$, which implies that the Bayes discrimination error decays exponentially in $H$ (up to constants in the exponent).*

*Proof sketch.* Factorize $P_\theta^\pi(\tau) = \prod_{t=1}^{H} \pi(a_t \mid h_{t-1}) p_\theta(o_t \mid a_t, h_{t-1})$ and note that policy terms cancel in the log-likelihood ratio. A KL chain rule decomposes the trajectory KL into a sum of per-round conditional KL terms, each lower bounded by $c$. A standard KL-to-testing bound yields exponential decay.

Let $L_{\text{stop}}(h) = \min_{\hat{y}} \mathbb{E}[\ell(\hat{y}) \mid h]$ be the minimal expected loss of stopping now, where $\ell$ is a task loss, and let $c(a)$ be the interaction cost. Define $\text{VoI}(h, a) = L_{\text{stop}}(h) - \mathbb{E}[L_{\text{stop}}(h \oplus (a, o)) \mid h, a]$.

**Theorem 4.4** (VoI-Cost Tradeoff). *The optimal policy terminates if and only if $\max_{a \in \mathcal{A}_{\text{inq}}} \text{VoI}(h, a) \leq \min_{a \in \mathcal{A}_{\text{inq}}} c(a)$.*

*Proof sketch.* View termination as an optimal stopping problem and compare stopping now versus one-step lookahead interaction via Bellman optimality.

The intrinsic curiosity drive in our AEI reward can be viewed as a proxy for VoI, encouraging evidence acquisition in histories where the expected information value is high, thus aligning exploration with rational termination. Complete proofs are provided in Appendix B.

## 5. Experiments

### 5.1. Experimental Setup

**Datasets and evaluation.** We evaluate Anom-$\pi$ on four benchmarks covering multi-scenario anomaly detection (XD-Violence (Wu et al., 2020), UCF-Crime (Sultani et al., 2018)), open-set generalization (UBnormal (Acsin-

*Table 1.* Frame-level performance (AUC/AP) on UCF-Crime, XD-Violence, UBnormal, and CSAD. Expl. indicates whether the method outputs an explanation. Reas. indicates whether explicit reasoning traces are used. Lrn. indicates whether a learned policy is used. Obs. indicates the evidence acquisition setting. An asterisk (*) indicates a closed-source commercial model.

| Methods | #Params | Expl. | Reas. | Lrn. | Obs. | Multi-Scenario | | Open-Set | Complex Scenario |
| | | | | | | UCF (AUC%) | XD (AP%) | UB (AUC%) | CSAD (AUC%) |
| --- | --- | --- | --- | --- | --- | --- | --- | --- | --- |
| *Passive Specialized Learning* | | | | | | | | | |
| UR-DMU (Zhou et al., 2023) (AAAI'23) | - | ✗ | ✗ | - | ✗ | 86.97 | 81.66 | 59.91 | - |
| AED-MAD (Ristea et al., 2024) (CVPR'24) | - | ✗ | ✗ | - | ✗ | - | - | 58.50 | - |
| VadCLIP (Wu et al., 2024) (AAAI'24) | - | ✗ | ✗ | - | ✗ | 88.02 | 84.51 | - | - |
| Ex-VAD (Huang et al., 2025) (ICML'24) | - | ✓ | ✗ | - | ✗ | 88.29 | 86.52 | - | - |
| π-VAD (Majhi et al., 2025) (CVPR'25) | - | ✗ | ✗ | - | ✗ | 90.33 | 85.37 | - | - |
| *Passive General Understanding* | | | | | | | | | |
| ZS CLIP (Radford et al., 2021) (ICML'21) | 0.3B | ✓ | ✗ | ✗ | Fixed | 53.16 | 17.83 | 46.20 | 32.45 |
| LLaVA-1.5 (Liu et al., 2024) (CVPR'24) | 13B | ✓ | ✗ | ✗ | Fixed | 72.84 | 50.26 | 53.71 | 47.78 |
| LAVAD (Zanella et al., 2024) (CVPR'24) | 13B | ✓ | ✗ | ✗ | Fixed | 80.28 | 62.01 | 64.23 | 57.26 |
| AnomalyRuler (Yang et al., 2024) (ECCV'24) | 17B* | ✓ | ✗ | ✓ | Fixed | - | - | 71.90 | - |
| VERA (Ye et al., 2025) (CVPR'25) | 8B | ✓ | ✗ | ✓ | Fixed | **86.55** | 70.11 | - | - |
| URF (Lin et al., 2026) (NeurIPS'25) | 7B | ✓ | ✗ | ✗ | Fixed | 84.28 | 68.07 | 69.02 | - |
| EventVAD (Shao et al., 2025) (ACM MM'25) | 7B | ✓ | ✗ | ✗ | Dyn. | 82.03 | 64.04 | - | - |
| VADTree (Li et al., 2026a) (NeurIPS'25) | 8B | ✓ | ✗ | ✗ | Dyn. | 84.74 | 68.85 | - | - |
| PANDA (Yang et al., 2026) (NeurIPS'25) | 72B* | ✓ | ✓ | ✗ | Dyn. | 84.89 | 70.16 | 75.78 | 73.12 |
| *Active Exploratory Reasoning* | | | | | | | | | |
| **Anom-π (ours)** | 2B | ✓ | ✓ | ✓ | OnD. | 84.46 | **72.29** | **78.75** | **79.18** |

toae et al., 2022)), and complex scenarios (CSAD (Yang et al., 2026)). We report frame-level ROC-AUC (%) on UCF-Crime/UBnormal/CSAD and frame-level AP(%) on XD-Violence. For each clip episode, Anom-π outputs the predicted anomaly category, temporal range, and confidence. For frame-level scoring, clips predicted as normal assign score 0 to all frames; clips predicted as anomalous assign score 1 inside the predicted anomalous range(s) and score 0.5 to the remaining frames in the same clip. When multiple clips overlap on the same video frame, we use the mean of their local scores as the final frame-level anomaly score.

**Implementation details.** Following prior work (Zanella et al., 2024; Ye et al., 2025), we uniformly subsample the raw video by taking one frame every 16 frames. We then form overlapping clips on the subsampled frame sequence with clip length $L=16$ and stride $\Delta=8$, both measured in subsampled frames, corresponding to a 50% overlap between adjacent clips. At inference time, Anom-π starts from 16 frames and can request at most 16 additional frames on demand. During deliberation (THINK), the model maintains up to three hypotheses $(\hat{y}_{i,m}, \hat{b}_{i,m})$ (anomaly category and temporal range within the clip). We use Qwen3-VL-Instruct-2B (Yang et al., 2025) as the backbone. Our agent operates over a unified action space that interleaves delib-

eration (THINK) with on-demand evidence inquiry actions (BACKTRACK, EXPAND, SAMPLE), followed by termination (FINAL). Unless otherwise stated, we use the same backbone across all ablations and only vary the available action set, whether iterative evidence update is allowed, and the policy learning objective. (Sec. 5.4) All experiments are conducted on two NVIDIA H800 GPUs using PyTorch.

## 5.2. Main Results

Table 1 summarizes frame-level performance on four benchmarks. On XD-Violence, Anom-π achieves 72.29% AP and outperforms PANDA (Yang et al., 2026) by 2.13% while using a much smaller backbone. On UBnormal, which emphasizes open-set generalization, Anom-π reaches 78.75% AUC and improves over UR-DMU (Zhou et al., 2023) by 18.84% and PANDA (Yang et al., 2026) by 2.97%. On CSAD, a complex-scenario benchmark, Anom-π attains 79.18% AUC and exceeds PANDA (Yang et al., 2026) by 6.06%. These results show that active on-demand evidence acquisition can compensate for smaller model scale, improve robustness under distribution shift, and support complex-scenario anomaly understanding.

These gains support our view of video anomaly understanding as closed-loop hypothesis verification. With only video-level labels, iDPO learns when to query, revise, and stop, improving evidence localization without dense temporal supervision. On UCF-Crime, Anom-π remains competitive with 84.46% AUC, although the smaller gain may reflect domain-specific scene patterns that are underrepresented in our training rollouts. The class-wise results in Appendix D.4 further show that interaction is most helpful for localized and subtle categories, where decisive cues are easy to miss

*Table 2.* Efficiency and paradigm comparison (Vid./Frm.: video-/frame-level on XD-Violence).

| Method | #Params | Data | Context | XD-V/AP(%) | |
| | | | | Vid. | Frm. |
| --- | --- | --- | --- | --- | --- |
| Holmes-VAU | 2B | 70k | 16 | 87.68 | – |
| VERA | 8B | ∼3.5k | 300 | – | 70.11 |
| **Anom-π (ours)** | 2B | ∼3.5k | 16–32 | **91.31** | **72.29** |

under a fixed initial view.

## 5.3. Efficiency and Paradigm Comparison

We summarize efficiency and paradigm differences in model/data scale and inference-time context, and relate them to localization performance. We compare against recent LLM-based baselines (e.g., Holmes-VAU (Zhang et al., 2025) and VERA (Ye et al., 2025)) in terms of backbone scale, weakly-supervised training data, and inference-time context budget. Holmes-VAU relies on a newly curated anomaly-understanding instruction dataset with roughly 70k annotated training samples. In contrast, like VERA, Anom-$\pi$ is trained directly on existing benchmarks using only their standard video-level labels (i.e., without extra manual annotation). Table 2 shows that VERA uses 300 frames, while Anom-$\pi$ uses only 16-32 frames. Despite the much smaller context, Anom-$\pi$ improves frame-level AP on XD-Violence from 70.11% (VERA) to 72.29%, and also achieves higher video-level AP (91.31% vs. 87.68% of Holmes-VAU).

## 5.4. Ablation Studies

In this section, we conduct extensive ablation studies to verify the effectiveness of our active policy design and the necessity of each component in Anom-$\pi$. All results are reported on the XD-Violence (Wu et al., 2020) benchmark.

*Table 3.* Ablation on interaction for closed-loop vs. one-shot pipeline.

| Method / Setting | AP(%) ↑ | AUC(%) ↑ |
|---|---|---|
| One-shot pipeline (random sampling) | 52.54 | 78.87 |
| One-shot pipeline (uniform sampling) | 65.99 | 89.48 |
| Active Interaction (Anom-$\pi$) | **72.29** | **93.16** |

**Effectiveness of active interaction.** To test whether the gain comes from closed-loop evidence acquisition rather than simply observing more frames, we compare Anom-$\pi$ with two One-shot Pipeline baselines that use either random or uniform frame sampling. Both baselines receive the initial 16 frames and 16 additional frames selected before reasoning starts, so they have a comparable frame budget but no intermediate hypothesis update. Random sampling reaches only 52.54 AP(%), indicating that sparse abnormal moments are easily missed when extra frames are not targeted. Uniform sampling improves coverage and reaches 65.99 AP(%), but it still trails Anom-$\pi$ at 72.29 AP(%) because the model cannot revise a hypothesis and query the uncertain temporal region. The comparison therefore isolates the value of the learned interaction policy, which decides where to query, when to revise, and when to stop under the same anomaly-understanding task.

**Policy learning objective.** Under the same backbone and action space, policy learning must decide not only which

*Table 4.* Ablation on the policy learning objective.

| Method | Interaction Rate | AP(%) ↑ | AUC(%) ↑ |
|---|---|---|---|
| ZERO-SHOT | 5.38% | 34.92 | 56.98 |
| SFT (positive-only) | 40.10% | 56.86 | 78.84 |
| DPO (w/o curiosity) | 36.50% | 66.57 | 91.14 |
| iDPO (ours) | 11.74% | **72.29** | **93.16** |

*Table 5.* Ablation on the action space (T: THINK, B: BACKTRACK, E: EXPAND, S: SAMPLE).

| Variant | T | B | E | S | AP(%) ↑ | AUC(%) ↑ |
|---|---|---|---|---|---|---|
| w/o THINK | ✗ | ✓ | ✓ | ✓ | 65.54 (-6.75) | 89.66 (-3.50) |
| w/o BACKTRACK | ✓ | ✗ | ✓ | ✓ | 70.49 (-1.80) | 92.74 (-0.42) |
| w/o EXPAND | ✓ | ✓ | ✗ | ✓ | 71.48 (-0.81) | 92.04 (-1.12) |
| w/o SAMPLE | ✓ | ✓ | ✓ | ✗ | 67.83 (-4.46) | 91.93 (-1.23) |
| FULL (Anom-$\pi$) | ✓ | ✓ | ✓ | ✓ | **72.29** | **93.16** |

anomaly label to predict, but also when additional evidence is worth acquiring and when the model should stop. As shown in Table 4, iDPO improves XD-Violence AP(%) from 66.57 to 72.29 over vanilla DPO (Rafailov et al., 2023) without AEI, indicating that trajectory preference alignment alone is insufficient for this interleaved decision process. The gain comes from AEI explicitly valuing informative, non-redundant evidence acquisition while penalizing unnecessary interaction, which encourages the policy to query under uncertainty and terminate once evidence is sufficient. A finer trajectory-construction ablation further shows why this cost-aware ranking is necessary. Scoring trajectories only by the final prediction drops AP by 5.42 points, while removing the interaction cost still loses 2.25 points relative to full iDPO (Appendix D.3). Detailed hyperparameter analyses are provided in Appendix D.2.

**Trajectory-level iDPO vs. GRPO.** GRPO (Guo et al., 2025) is a strong and complementary optimization family, and recent anomaly reasoning works further demonstrate the value of reinforcement-style reasoning optimization (Huang et al., 2026; Zhu et al., 2025; Mo et al., 2026; Li et al., 2026b; Yu et al., 2026; Kang et al., 2026). The key difference lies in the supervision interface. In weakly supervised VAU, two interleaved trajectories can reach the same final video-level label while differing in evidence quality, redundancy, hypothesis verification, and stopping. iDPO directly aligns AEI-ranked chosen/rejected trajectories, thus optimizing the behavior we care about, namely well-justified evidence acquisition and rational termination, rather than only the final label. Applying GRPO to these long rollouts would require converting trajectory-level scores into scalar rewards and group-relative advantages, which is feasible but more sensitive to reward noise and credit assignment. It is also more expensive in our setting. On the same H800 setup, a GRPO-style rollout optimization requires about 140 GB peak VRAM and over 34 hours per epoch, compared with

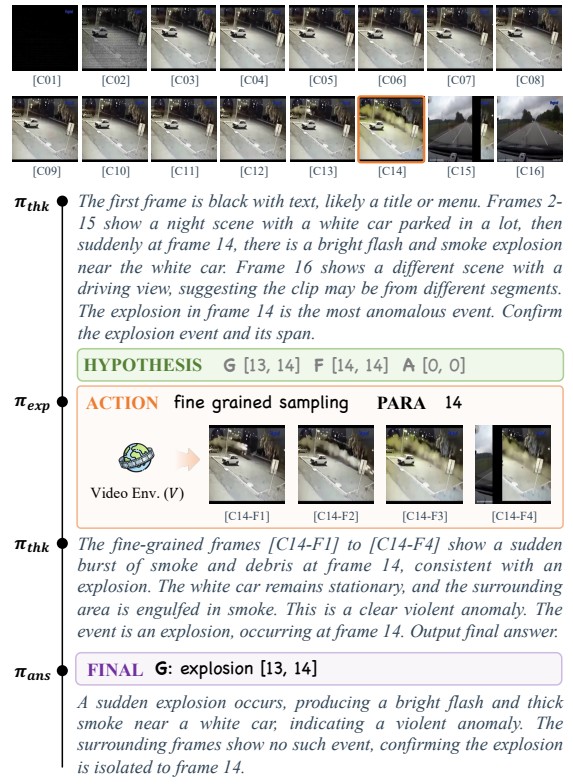

$\pi_{thk}$ *The first frame is black with text, likely a title or menu. Frames 2-15 show a night scene with a white car parked in a lot, then suddenly at frame 14, there is a bright flash and smoke explosion near the white car. Frame 16 shows a different scene with a driving view, suggesting the clip may be from different segments. The explosion in frame 14 is the most anomalous event. Confirm the explosion event and its span.*

**HYPOTHESIS**  G [13, 14]  F [14, 14]  A [0, 0]

$\pi_{exp}$  **ACTION**  fine grained sampling  **PARA**  14

Video Env. (V)

[C14-F1]  [C14-F2]  [C14-F3]  [C14-F4]

$\pi_{thk}$ *The fine-grained frames [C14-F1] to [C14-F4] show a sudden burst of smoke and debris at frame 14, consistent with an explosion. The white car remains stationary, and the surrounding area is engulfed in smoke. This is a clear violent anomaly. The event is an explosion, occurring at frame 14. Output final answer.*

$\pi_{ans}$  **FINAL**  G: explosion [13, 14]

*A sudden explosion occurs, producing a bright flash and thick smoke near a white car, indicating a violent anomaly. The surrounding frames show no such event, confirming the explosion is isolated to frame 14.*

*Figure 3.* **Instantaneous anomaly case.** Anom-$\pi$ refines temporal boundaries by probing around a suspected moment and then terminates early.

about 54 GB and 2 hours for iDPO. We therefore view GRPO as complementary rather than inferior, and leave GRPO training with richer rewards as future work.

**Action space analysis.** We evaluate the contribution of different actions by removing them from the action space $\mathcal{A}$. Without BACKTRACK, performance degrades on short, bursty anomalies, since the model cannot refine temporal boundaries once the observation window moves forward. Without EXPAND/SAMPLE, restricting the model to a fixed temporal resolution limits its ability to disambiguate fine-grained events (e.g., distinguishing "explosion" from "car accident"), leading to a noticeable drop in detection performance. Overall, these results highlight that a unified, interleaved action space is vital for handling the diverse nature of video anomalies.

**Behavioral diagnostics.** Beyond final accuracy, we examine whether the policy interacts for the right reason rather than simply requesting more frames. We use Evidence Refinement Rate (ERR) to capture additional evidence seeking, Hypothesis Revision Rate (HRR) to capture belief updates after observation, and premature finalization rate to capture early stopping errors. The diagnostics in Fig. 5 show that iDPO attains the strongest HRR and substantially reduces premature finalization, while not maximizing ERR. This

*Table 6.* Amortized interaction overhead of Anom-$\pi$ at inference time. Avg. Steps counts evidence-acquisition actions beyond the initial observation; Extra Frames denotes additionally queried frames per clip.

| Dataset | Avg. Steps | Extra Frames | Latency (s) |
|---|---|---|---|
| XD-Violence | 0.265 | 1.915 | 2.503 |
| UCF-Crime | 0.246 | 1.489 | 2.077 |
| UBnormal | 0.180 | 1.011 | 1.492 |
| CSAD | 0.397 | 3.826 | 3.877 |

*Table 7.* Backbone generalization on XD-Violence. Base denotes the corresponding backbone without the learned Anom-$\pi$ interaction policy.

| Backbone | Base | | Anom-$\pi$ (Ours) | |
|---|---|---|---|---|
| | AP(%) | AUC(%) | AP(%) | AUC(%) |
| Qwen3-VL-8B-Instruct | 58.83 | 82.35 | 74.46 (+15.63) | 93.47 (+11.12) |
| InternVL3.5-2B-Instruct | 44.78 | 72.34 | 66.18 (+21.40) | 89.24 (+16.90) |

pattern suggests more targeted refinement: the policy tends to interact when new evidence is likely to change the hypothesis, instead of following a fixed exploration budget. Appendix D.1 provides complementary action-distribution and revision statistics, further showing that interaction is concentrated on ambiguous clips.

**Interaction overhead analysis.** Table 6 quantifies the amortized inference overhead introduced by selective interaction. On XD-Violence, Anom-$\pi$ requests 0.265 evidence-acquisition steps and 1.915 extra frames per clip on average, resulting in 2.503s latency. Across datasets, the average number of interaction steps remains below 0.4, indicating that most clips are resolved after the initial observation. This latency is lower than VERA at 2.882s and LAVAD at 3.912s on XD-Violence, even though Anom-$\pi$ performs closed-loop reasoning. These results show that Anom-$\pi$ improves localization while keeping inference overhead controlled. We do not claim hard real-time deployment. Appendix D.1 provides the full dataset-level overhead breakdown and latency comparison details.

**Backbone generalization analysis.** To examine whether the learned interaction policy is tied to the default 2B Qwen backbone, we evaluate it with two additional VLMs. With Qwen3-VL-8B-Instruct (Yang et al., 2025) and InternVL3.5-2B-Instruct (Wang et al., 2025), Anom-$\pi$ raises XD-Violence AP(%) from 58.83 to 74.46 and from 44.78 to 66.18, respectively (Table 7). These consistent improvements indicate that the policy is not merely exploiting idiosyncrasies of the default backbone. The larger gain on InternVL3.5 further suggests that active evidence acquisition is especially helpful when the base model is weaker, whereas the improvement with Qwen3-VL-8B shows that the same interaction mechanism remains beneficial at a larger model scale.

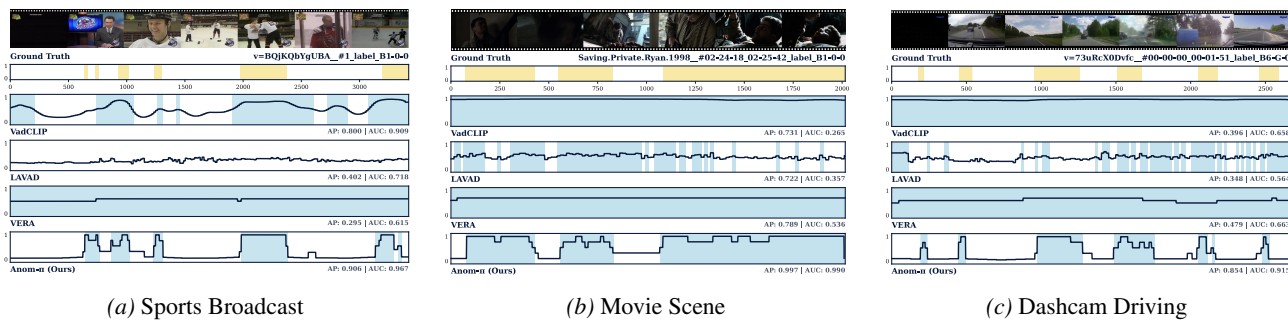

*(a)* Sports Broadcast        *(b)* Movie Scene        *(c)* Dashcam Driving

*Figure 4.* **Temporal anomaly score curves.** We compare Anom-$\pi$ with VadCLIP, LAVAD, and VERA on three scenarios. Yellow indicates the ground-truth anomalous interval and blue indicates predicted anomalous spans after thresholding the anomaly score at 0.5.

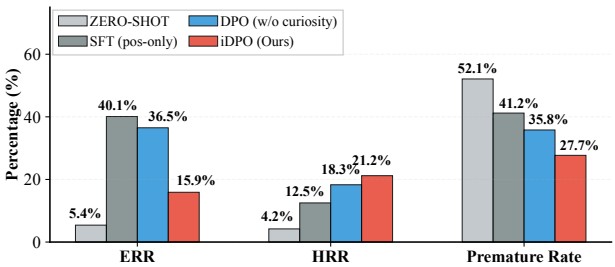

*Figure 5.* **Behavioral diagnostics.** Evidence Refinement Rate (ERR), Hypothesis Revision Rate (HRR), and premature finalization rate under different training objectives.

### 5.5. Visualization Analysis

**Active interaction verifies subtle anomalies.** The example in Fig. 3 is difficult because the anomaly is almost instantaneous: the first frame is a black title-like frame, frames 2-13 show a stationary white car at night, and only frame 14 contains a brief flash with smoke near the car. Under the coarse 16-frame view, this cue can be diluted by neighboring normal frames or confused with a traffic accident. Anom-$\pi$ therefore narrows the hypothesis to the suspected timestamp and invokes fine-grained sampling around frame 14. The returned subframes reveal smoke and debris while the car remains stationary, supporting an Explosion rather than a road-accident interpretation. Since frame 16 has already switched to a different driving scene, the model also uses the queried evidence to bound the event tightly and terminates with FINAL instead of continuing redundant exploration.

**Temporal anomaly score curves compared with prior methods.** The examples in Fig. 4 show how Anom-$\pi$ handles different temporal structures and visual distractors. In the sports broadcast clip, logos, commentator shots, player close-ups, normal skating, and hockey fights appear in rapid alternation. VadCLIP (Wu et al., 2024) and VERA (Ye et al., 2025) respond to broad visually related regions, while LAVAD (Zanella et al., 2024) stays nearly flat, but Anom-$\pi$ produces separated peaks around the annotated fight inter-

vals and suppresses most broadcast cutaways. In the movie clip, close-up combat scenes form several long abnormal spans separated by quieter transitions, where Anom-$\pi$ preserves the normal gap more clearly than baselines that saturate or fragment the timeline. In the dashcam clip, short accident-related cues are embedded in long normal driving views such as open roads, passing cars, and a blue truck. Anom-$\pi$ recovers more localized peaks around these sparse intervals, suggesting that closed-loop evidence acquisition helps distinguish brief abnormal evidence from visually similar background.

### 5.6. Limitations

The above results show the benefit of moving beyond a fixed observation pipeline, as Anom-$\pi$ can request evidence and stop adaptively as its hypothesis evolves. Its remaining limitations mainly arise when the returned evidence is still intrinsically ambiguous. Low-quality scenes may induce redundant exploration, and overly local inspection may lead to premature stopping. We discuss representative cases in Appendix E. Future policies could better calibrate uncertainty before FINAL and estimate evidence value under poor visibility or motion blur.

### 6. Conclusion

We introduced Anom-$\pi$, a video anomaly understanding framework that formulates inference as *active* hypothesis verification with interleaved reasoning and observation. Starting from a short context, Anom-$\pi$ alternates THINK with adaptive observation and learns when to stop with FINAL. To train this policy under weak supervision, iDPO directly optimizes AEI-induced trajectory preferences. Across benchmarks, Anom-$\pi$ achieves strong frame-level performance with compact inference-time context, while ablations confirm the importance of interaction and iDPO for localizing short and subtle anomalies. Overall, Anom-$\pi$ shows that jointly optimizing reasoning and observation provides an effective active paradigm for video anomaly understanding.

## Acknowledgements

This work was supported in part by the New Generation Artificial Intelligence-National Science and Technology Major Project under Grant No. 2025ZD0123601, in part by the National Natural Science Foundation of China under Grants No. 62472060 and 62221005, in part by the Science and Technology Innovation Key R&D Program of Chongqing under Grant No. CSTB2023TIAD-STX0016, in part by the Natural Science Foundation of Chongqing under Grants No. CSTB2024NSCQ-QCXMX0060, in part by the China Postdoctoral Science Foundation under Grant No. 2025MD774186, in part by the Chongqing Special Postdoctoral Research Funding under Grant No. 2024CQB-SHTB2002, in part by the Chongqing University of Posts and Telecommunications Ph.D. Innovative Talents Project under Grant No. BYJS202404.

## Impact Statement

This work advances video anomaly understanding by shifting from passive inference to active, evidence-driven inquiry. A key positive impact is improved interpretability and reliability in safety monitoring. By generating natural language explanations and grounding predictions in the evidence-acquisition process, the model can provide human operators with more context for its decisions. In addition, the rational termination mechanism can improve computational efficiency by reducing redundant processing in simple scenarios. Potential risks include privacy infringement and biased detection in diverse environments. To mitigate these risks, we advocate deployment with human-in-the-loop verification, rigorous oversight of the model outputs, and privacy-preserving data handling.

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

## A. Additional Theoretical Analysis Details

### A.1. Theoretical setup and paradigm shift

We interpret curiosity as an intrinsic drive that allocates interaction to evidence-deficient states, encouraging the agent to acquire information when its current explanation is insufficient.

**Remark.** From a physics and control perspective, curiosity corresponds to maximizing expected information value; equivalently, it reduces a free-energy-like objective under an interaction cost, yielding principled interaction and termination behavior.

We analyze a paradigm shift. Traditional video anomaly understanding follows *open-loop* observation followed by *cognition-only* reasoning. In contrast, Anom-$\pi$ follows *closed-loop* interleaved inference, where reasoning and interaction are interwoven.

### A.2. Interleaved inference with termination (formalization)

Let $\theta \in \Theta$ denote the latent anomaly explanation and let $X \sim p_\theta$ be the underlying video process. The agent interacts for at most $T$ rounds. At round $t$, it chooses an action $a_t \in \mathcal{A}$, where $\mathcal{A}$ includes interaction actions and a termination action $\text{FINAL}(\hat{y})$. If $a_t$ is an interaction action, the environment returns an observation $o_t$ and the history is updated. If $a_t = \text{FINAL}(\hat{y})$, the episode terminates and outputs $\hat{y}$:

$$
\begin{aligned}
a_t &\sim \pi(\cdot \mid h_{t-1}), \\
o_t &\sim p_\theta(\cdot \mid a_t, h_{t-1}), \\
h_t &= h_{t-1} \oplus (a_t, o_t),
\end{aligned}
\tag{7}
$$

where $\oplus$ denotes concatenation. Termination is therefore modeled as a decision of the policy, rather than an external thresholding rule.

### A.3. Sufficient condition for effective closed-loop policies

Assumption 4.2 in the main text is existential and is not a property guaranteed by arbitrary interaction. One concrete sufficient condition for the assumption is the following. Define the per-action conditional KL gap at round $t$ as

$$
\Delta_t(h_{t-1}, a) = D_{\text{KL}}\big(p_\theta(\cdot \mid a, h_{t-1}) \,\|\, p_{\theta'}(\cdot \mid a, h_{t-1})\big).
\tag{8}
$$

Suppose that for each round $t$ and for $P_\theta^\pi$-almost every history $h_{t-1}$, there exists an informative action set $\mathcal{A}_{\text{info}}(h_{t-1}) \subseteq \mathcal{A}$ such that

$$
\Delta_t(h_{t-1}, a) \geq \delta \quad \forall a \in \mathcal{A}_{\text{info}}(h_{t-1}), \qquad \pi\big(\mathcal{A}_{\text{info}}(h_{t-1}) \mid h_{t-1}\big) \geq \rho,
\tag{9}
$$

for constants $\rho > 0$ and $\delta > 0$. Since $\Delta_t(h_{t-1}, a) \geq 0$, the expected per-round discriminability gain is lower bounded as

$$
\begin{aligned}
&\mathbb{E}_{h_{t-1}, a_t \sim P_\theta^\pi}[\Delta_t(h_{t-1}, a_t)] \\
&\geq \delta \, \mathbb{E}_{h_{t-1} \sim P_\theta^\pi}\big[\pi\big(\mathcal{A}_{\text{info}}(h_{t-1}) \mid h_{t-1}\big)\big] \geq \rho\delta.
\end{aligned}
\tag{10}
$$

Thus Assumption 4.2 holds with $c = \rho\delta$ under this sufficient condition.

In practice, AEI does not explicitly estimate $\Delta_t$, $\rho$, or $\delta$, and iDPO does not formally enforce the above condition. Instead, $r_{\text{inf}}$ rewards non-redundant evidence queries, $r_{\text{hyp}}$ rewards hypothesis refinement, and the cost term discourages redundant probing; iDPO then prefers trajectories with higher AEI utility. These signals are empirical proxies aligned with effective evidence acquisition, not a formal guarantee that every learned policy satisfies Assumption 4.2.

## B. Proofs for Theoretical Analysis

This appendix (Appendix B) collects complete proofs for the theoretical statements in Sec. 4.

### B.1. Proof of Theorem 4.3

Fix $\theta \neq \theta'$ and a closed-loop policy $\pi$. Let $P_\theta^\pi$ denote the trajectory distribution of $\tau = (a_{1:T}, o_{1:T})$ induced by $\pi$ and the environment conditionals $p_\theta(o_t \mid a_t, h_{t-1})$. Because the policy is fixed, the action conditionals $\pi(a_t \mid h_{t-1})$ are shared under both $\theta$ and $\theta'$. Thus we can factor the trajectory distributions as

$$P_\theta^\pi(\tau) = \prod_{t=1}^T \pi(a_t \mid h_{t-1})\, p_\theta(o_t \mid a_t, h_{t-1}), \qquad P_{\theta'}^\pi(\tau) = \prod_{t=1}^T \pi(a_t \mid h_{t-1})\, p_{\theta'}(o_t \mid a_t, h_{t-1}). \tag{11}$$

**KL chain rule along the interaction.** Using the above factorization, we expand the log-likelihood ratio as

$$\log \frac{P_\theta^\pi(\tau)}{P_{\theta'}^\pi(\tau)} = \sum_{t=1}^T \log \frac{p_\theta(o_t \mid a_t, h_{t-1})}{p_{\theta'}(o_t \mid a_t, h_{t-1})}, \tag{12}$$

since the policy terms cancel. Taking expectation under $\tau \sim P_\theta^\pi$ yields

$$\begin{aligned} D_{\mathrm{KL}}\big(P_\theta^\pi \,\|\, P_{\theta'}^\pi\big) &= \mathbb{E}_{\tau \sim P_\theta^\pi}\left[\log \frac{P_\theta^\pi(\tau)}{P_{\theta'}^\pi(\tau)}\right] = \sum_{t=1}^T \mathbb{E}_{\tau \sim P_\theta^\pi}\left[\log \frac{p_\theta(o_t \mid a_t, h_{t-1})}{p_{\theta'}(o_t \mid a_t, h_{t-1})}\right] \\ &= \sum_{t=1}^T \mathbb{E}_{h_{t-1}, a_t \sim P_\theta^\pi}\big[D_{\mathrm{KL}}\big(p_\theta(\cdot \mid a_t, h_{t-1}) \,\|\, p_{\theta'}(\cdot \mid a_t, h_{t-1})\big)\big]. \end{aligned} \tag{13}$$

By Assumption (Per-round discriminability gain), each summand is lower bounded by $c$, hence

$$D_{\mathrm{KL}}\big(P_\theta^\pi \,\|\, P_{\theta'}^\pi\big) \geq cT. \tag{14}$$

**From KL separability to exponential testing error.** For binary testing between $P_\theta^\pi$ and $P_{\theta'}^\pi$ with equal priors, let $P_e^\pi(T)$ denote the minimal (Bayes) error probability. A standard inequality (Bretagnolle-Huber) gives

$$P_e^\pi(T) \leq \tfrac{1}{2} \exp\left(-\tfrac{1}{2} D_{\mathrm{KL}}\big(P_\theta^\pi \,\|\, P_{\theta'}^\pi\big)\right). \tag{15}$$

Combining with $D_{\mathrm{KL}}(P_\theta^\pi \,\|\, P_{\theta'}^\pi) \geq cT$ yields

$$P_e^\pi(T) \leq \tfrac{1}{2} \exp\left(-\tfrac{c}{2}T\right), \tag{16}$$

which establishes an exponential decay in $T$ (up to constant factors in the exponent).

### B.2. Proof of Theorem 4.1

Fix $\theta \neq \theta'$ satisfying $D_{\mathrm{TV}}(p_\theta^g, p_{\theta'}^g) \leq \varepsilon$. Consider testing $H_0 : \vartheta = \theta$ versus $H_1 : \vartheta = \theta'$ based on $Y \sim p_\vartheta^g$. By Le Cam's two-point method, for any estimator $\hat{\theta}(Y)$,

$$\inf_{\hat{\theta}} \sup_{\vartheta \in \{\theta, \theta'\}} \Pr_\vartheta\big(\hat{\theta}(Y) \neq \vartheta\big) \geq \tfrac{1}{2}\big(1 - D_{\mathrm{TV}}(p_\theta^g, p_{\theta'}^g)\big) \geq \tfrac{1}{2}(1 - \varepsilon), \tag{17}$$

which proves the claim.

### B.3. Proof of Theorem 4.4

At history $h$, stopping incurs cost $L_{\mathrm{stop}}(h)$. Taking one additional interaction $a$ incurs the immediate interaction cost (as defined in the theorem) plus the expected post-interaction stopping cost $\mathbb{E}[L_{\mathrm{stop}}(h \oplus (a, o)) \mid h, a]$. Define

$$L_{\mathrm{ask}}(h, a) := \text{interaction-cost}(h, a) + \mathbb{E}\big[L_{\mathrm{stop}}(h \oplus (a, o)) \mid h, a\big]. \tag{18}$$

The optimal one-step decision is to terminate iff $L_{\mathrm{stop}}(h) \leq \min_{a \in \mathcal{A}} L_{\mathrm{ask}}(h, a)$, which is exactly the value-of-information stopping criterion.

# C. Interaction Tools: Design and Interfaces

## C.1. Tool taxonomy and design principles

We design tools as a minimal set of *atomic* evidence query operators that correspond to common human video reviewing behaviors, while keeping the interface simple and reproducible. Our tool set decomposes into (i) a deliberation action (THINK) with no environment side effects; (ii) investigative actions (BACKTRACK, EXPAND, SAMPLE) that change the observation view and return new evidence; and (iii) a termination action (FINAL) that outputs the final structured prediction.

Each tool follows the same schema constrained calling convention, which standardizes invocation and error handling. We impose lightweight constraints on parameters (e.g., bounded indices and limited return sizes) to control latency and prevent runaway interaction, and we normalize tool outputs into a unified return structure (`status`, `new_image_paths`, `followup_query`) so that downstream reasoning can remain tool agnostic.

## C.2. Tool specifications

We use the OpenAI HERMES tool-calling format, and specify each tool interface as a compact, machine-readable schema (JSON Schema / function-calling schema), accompanied by a short natural-language contract. In the main paper we summarize the tool set at a high level; here we provide the full schema for reproducibility.

**Tool: THINK (`think`).** **Purpose:** record private analysis/planning to decide the next action; *no side effects*. **I/O:** input is a JSON object satisfying the schema below; output returns `next_action` and a `followup_query` string.

```
{
  "type": "function",
  "function": {
    "name": "think",
    "description": "Record private analysis/planning before the next action. Include the
        reason for choosing the next tool inside 'thought'.",
    "parameters": {
      "type": "object",
      "properties": {
        "thought": {
          "type": "string",
          "description": "Internal analysis + decision, keep brief. Format: 'OBS: ...;
              DECIDE: ...; PLAN: ...'."
        },
        "hyp": {
          "type": "object",
          "description": "Current candidate hypotheses before taking the next action.",
          "properties": {
            "status": {"type": "string", "enum": ["unknown", "tentative", "confident"]},
            "cands": {
              "type": "array",
              "minItems": 0,
              "maxItems": 3,
              "description": "Up to 3 candidates ordered by confidence (highest first).",
              "items": {
                "type": "object",
                "properties": {
                  "c": {
                    "type": "array",
                    "minItems": 1,
                    "maxItems": 1,
                    "items": {"type": "string"},
                    "description": "Single category code. Use ['A'] for Normal."
                  },
                  "anom": {
                    "type": "array",
                    "minItems": 2,
                    "maxItems": 2,
```

```
                        "items": {"type": "integer", "minimum": 0, "maximum": 16},
                        "description": "Anomaly span within the 16-frame input, 1-based inclusive
                            (1..16). Use [0,0] for Normal."
                    }
                },
                "required": ["c", "anom"],
                "additionalProperties": false
            }
        }
    },
    "required": ["status", "cands"],
    "additionalProperties": false
},
"next_action": {
    "type": "string",
    "description": "Next tool to call.",
    "enum": ["think", "get_segment_context", "select_frames", "fine_grained_sampling
        ", "final_answer"]
}
},
"required": ["thought", "hyp", "next_action"],
"additionalProperties": false
    }
  }
}
```

**Minimal example call.**

```
{"thought":"OBS: ...; DECIDE: sample frames; PLAN: ...",
 "hyp":{"status":"tentative","cands":[{"c":["B"],"anom":[5,10]}]},
 "next_action":"select_frames"}
```

**Tool: BACKTRACK (`select_frames`). Purpose:** select a small set of keyframes for local, fine-grained inspection when anomalies are temporally sparse. **I/O:** input specifies up to 4 indices within the 16-frame clip; output provides the selected frame paths and sets next_action=think.

```
{
  "type": "function",
  "function": {
    "name": "select_frames",
    "description": "Selects keyframes with potential abnormal activities. Useful when
        anomalies are temporally sparse or occur in short bursts. Maximum 4 frames
        allowed.",
    "parameters": {
      "type": "object",
      "properties": {
        "target_frames": {
          "type": "array",
          "description": "The target frame indices to select from the given frames.",
          "items": {
            "type": "integer",
            "description": "Frame index from 1 to 16.",
            "minimum": 1,
            "maximum": 16
          },
          "maxItems": 4
        }
      },
      "required": ["target_frames"],
      "additionalProperties": false
    }
```

```
   }
}
```

**Minimal example call.**

```
{"target_frames":[2,5,10,15]}
```

**Tool: EXPAND (`get_segment_context`). Purpose:** retrieve a small number of context frames immediately before/after the current segment. **I/O:** input specifies the side (`prev` or `next`); output returns `new_image_paths` and sets `next_action=think`.

```
{
  "type": "function",
  "function": {
    "name": "get_segment_context",
    "description": "Get context frames outside the provided segment (prev: before the
        provided segment; next: after the provided segment).",
    "parameters": {
      "type": "object",
      "properties": {
        "side": {"type": "string", "enum": ["prev", "next"]}
      },
      "required": ["side"],
      "additionalProperties": false
    }
  }
}
```

**Minimal example call.**

```
{"side":"prev"}
```

**Tool: SAMPLE (`fine_grained_sampling`). Purpose:** sample additional frames around a selected coarse frame to capture subtle / transient anomalies. **I/O:** input specifies a `target_frame` (1–16) within the current 16-frame clip; output returns up to 4 fine-grained frames around that coarse frame.

```
{
  "type": "function",
  "function": {
    "name": "fine_grained_sampling",
    "description": "Perform fine-grained frame sampling around a chosen target frame
        (1-16) to capture subtle and transient anomalies that may be missed by coarse
        sampling. Returns up to 4 fine-grained frames around the target.",
    "parameters": {
      "type": "object",
      "properties": {
        "target_frame": {
          "type": "integer",
          "description": "Target frame index (1 to 16)",
          "minimum": 1,
          "maximum": 16
        }
      },
      "required": ["target_frame"],
      "additionalProperties": false
    }
  }
}
```

**Minimal example call.**

```
{"target_frame":5}
```

**Tool: FINAL (`final_answer`). Purpose:** return the final prediction as a category code, anomaly span within the 16-frame input (1-based), and a brief explanation. **I/O:** input is `null`; output returns `c`, `anom`, and `exp` as the final structured prediction.

```
{
  "type": "function",
  "function": {
   "name": "final_answer",
   "description": "Return category code, anomaly span within the 16-frame input (1-based
       ), and a brief explanation.",
   "parameters": {
    "type": "object",
    "properties": {
     "c": {
       "type": "array",
       "minItems": 1,
       "maxItems": 1,
       "items": {"type": "string"},
       "description": "Single category code. Use 'A' for Normal."
     },
     "anom": {
       "type": "array",
       "minItems": 2,
       "maxItems": 2,
       "items": {"type": "integer", "minimum": 0, "maximum": 16},
       "description": "Anomaly span within the 16-frame input, 1-based inclusive
           (1..16). Use [0,0] for Normal."
     },
     "exp": {
       "type": "string",
       "description": "Brief explanation (1-2 sentences) with key visual cues."
     }
    },
    "required": ["c", "anom", "exp"],
    "additionalProperties": false
   }
  }
}
```

**Minimal example call.**

```
{"c":["A"],"anom":[0,0],"exp":"No anomalous activity is observed."}
```

**Discussion: unified tool/action format.** Our tool interfaces adopt a single, unified structured format across all actions (THINK, BACKTRACK, EXPAND, SAMPLE, and FINAL). This design makes the interaction protocol easy to learn and imitate during alignment: the model always emits a tool call with an explicit name and a schema constrained argument object, which significantly reduces format errors (e.g., malformed JSON, missing fields, or mixed templates) compared to ad hoc prompting.

More importantly, the unified format places *all* model decisions (deliberation, evidence acquisition, and termination) into the same discrete action space $\mathcal{A}$. This allows us to optimize a single interleaved policy $\pi$ over heterogeneous behaviors, enables direct preference comparisons between trajectories that differ in both *what* evidence is acquired and *when* to stop, and supports principled cost/utility tradeoffs for rational interaction.

# D. Additional Experimental Analyses

This section provides supporting experimental statistics that complement the main-text results without adding extra figures to the main paper. The emphasis is on whether the learned policy actually behaves like an adaptive reviewer: it should request extra evidence only when useful, revise hypotheses when newly acquired evidence changes the current belief, and remain stable under reasonable choices of preference-construction and evaluation hyperparameters.

*Table 8.* Dataset-level amortized interaction overhead of Anom-$\pi$. Each clip starts from 16 frames and can request up to 16 additional frames; Avg. Observed Frames is computed as 16 plus the average additionally queried frames.

| Dataset | Initial Frames | Avg. Steps | Extra Frames | Avg. Observed Frames | Latency (s) |
|---|---|---|---|---|---|
| XD-Violence | 16 | 0.265 | 1.915 | 17.915 | 2.503 |
| UCF-Crime | 16 | 0.246 | 1.489 | 17.489 | 2.077 |
| UBnormal | 16 | 0.180 | 1.011 | 17.011 | 1.492 |
| CSAD | 16 | 0.397 | 3.826 | 19.826 | 3.877 |

## D.1. Efficiency and Interaction Statistics

To make the overhead concrete, Table 8 reports amortized interaction costs across datasets. Starting from 16 initial frames, Anom-$\pi$ only requests 1.011-3.826 additional frames on average, so the actually observed context remains close to the initial view: 17.011 frames on UBnormal, 17.489 on UCF-Crime, 17.915 on XD-Violence, and 19.826 on CSAD. The average number of evidence-acquisition steps is also below one on all datasets (0.180-0.397), indicating that the policy does not spend a fixed multi-step budget on every clip. Instead, the overhead increases with apparent ambiguity: UBnormal has the lowest cost (0.180 steps, 1.011 extra frames, 1.492s), whereas CSAD requires the most additional evidence (0.397 steps, 3.826 extra frames, 3.877s).

*Table 9.* Per-clip latency comparison on XD-Violence. Lower latency is better; $\Delta$ reports the absolute latency gap relative to Anom-$\pi$.

| Method | Inference Context | Latency (s) | $\Delta$ vs. Anom-$\pi$ |
|---|---|---|---|
| LAVAD | – | 3.912 | +1.409 |
| VERA | 300 frames | 2.882 | +0.379 |
| Anom-$\pi$ | 16 initial + 1.915 queried frames (avg.) | **2.503** | 0.000 |

On XD-Violence, selective interaction also keeps latency competitive (Table 9). Although Anom-$\pi$ performs closed-loop reasoning, its 2.503s per-clip latency is lower than VERA (2.882s) and LAVAD (3.912s), because most clips are handled after the initial observation and only ambiguous clips invoke additional tools. This result should be interpreted as an amortized efficiency claim rather than a hard real-time guarantee.

*Table 10.* Action distribution and amortized interaction statistics across datasets. Avg. Steps counts evidence-acquisition actions beyond the initial observation, and Extra Frames denotes additionally queried frames per clip.

| Dataset | Backtrack (%) | Expand (%) | Sample (%) | Avg. Steps | Distinct Locs | Extra Frames | Latency (s) |
|---|---|---|---|---|---|---|---|
| XD-Violence | 10.33 | 14.22 | 75.46 | 0.265 | 1.857 | 1.915 | 2.503 |
| UCF-Crime | 41.71 | 12.44 | 45.85 | 0.246 | 1.334 | 1.489 | 2.077 |
| UBnormal | 25.92 | 21.14 | 52.94 | 0.180 | 0.852 | 1.011 | 1.492 |
| CSAD | 15.77 | 20.49 | 63.73 | 0.397 | 3.426 | 3.826 | 3.877 |

Beyond aggregate cost, Tables 10 and 11 separate action choices from hypothesis revisions. The action distribution is dataset-dependent: SAMPLE dominates XD-Violence, UBnormal, and CSAD (75.46%, 52.94%, and 63.73%), consistent with the need to verify transient local cues, while UCF-Crime uses much more BACKTRACK (41.71%), matching its longer surveillance-style videos where preceding context is often informative. Hypothesis revision is similarly concentrated on the subset of clips that actually trigger interaction. For example, only 2.49% of all XD-Violence clips revise their hypothesis, but 21.20% of the interacted clips do; on CSAD, the corresponding rates are 11.72% overall and 31.31% within interacted clips. Thus, revision is not a ubiquitous artifact of the tool format, but mainly appears when the policy has decided that extra evidence is worth acquiring.

*Table 11.* Hypothesis revision behavior across datasets. Inter. denotes clips with evidence acquisition beyond the initial observation; Rev. Overall reports the revision rate over all clips, and Rev. Within Inter. reports the revision rate among interacted clips.

| Dataset | Inter. (%) | Rev. Overall (%) | Rev. Within Inter. (%) |
|---|---|---|---|
| XD-Violence | 11.74 | 2.49 | 21.20 |
| UCF-Crime | 16.52 | 3.32 | 20.10 |
| UBnormal | 12.10 | 0.69 | 5.72 |
| CSAD | 37.43 | 11.72 | 31.31 |

*Table 12.* Hyperparameter sensitivity. Panel (a) reports the $\eta$ sweep with both XD-Violence AP and UCF-Crime AUC; Panel (b) reports $\lambda$ and $N_{\mathrm{buf}}$ sweeps on XD-Violence. The default setting is $\eta=0.5$, $\lambda=1.0$, and $N_{\mathrm{buf}}=2$.

**(a) Interaction-cost sweep.**

| $\eta$ | Inter. (%) | Avg. Steps | Dist. Locs | HRR (%) | XD AP (%) | UCF AUC (%) |
|---|---|---|---|---|---|---|
| 0.0 | 36.11 | 0.2865 | 1.8696 | 2.81 | 70.04 | 84.52 |
| 0.1 | 13.26 | 0.2658 | 1.8631 | 2.61 | 72.43 | 83.22 |
| 0.5 | 11.74 | 0.2650 | 1.8570 | 2.49 | 72.29 | 84.46 |
| 1.0 | 11.26 | 0.2593 | 1.7809 | 1.78 | 70.87 | 82.87 |
| 2.0 | 9.75 | 0.2439 | 1.5764 | 1.53 | 69.90 | 82.53 |

**(b) AEI-weight and buffer-size sweeps.**

| Hyperparam. | Value | Inter. (%) | Avg. Steps | Dist. Locs | HRR (%) | XD AP (%) |
|---|---|---|---|---|---|---|
| | 0.5 | 10.66 | 0.2507 | 1.6820 | 2.37 | 71.98 |
| $\lambda$ | 1.0 | 11.74 | 0.2650 | 1.8570 | 2.49 | 72.29 |
| | 2.0 | 12.59 | 0.2761 | 1.9913 | 2.68 | 72.15 |
| | 1 | 6.84 | 0.1819 | 1.0809 | 2.48 | 70.29 |
| $N_{\mathrm{buf}}$ | 2 | 11.74 | 0.2650 | 1.8570 | 2.49 | 72.29 |
| | 3 | 12.16 | 0.2683 | 1.8939 | 2.65 | 72.33 |

## D.2. Hyperparameter Sensitivity

For hyperparameters, we vary the interaction-cost weight $\eta$, the AEI weight $\lambda$, and the evidence buffer size $N_{\mathrm{buf}}$; the full sweep is given in Table 12. The interaction-cost sweep shows the clearest trade-off. When $\eta=0$, the policy explores much more frequently (36.11% interacted clips), but XD-Violence AP drops to 70.04, suggesting that removing the cost term encourages redundant evidence gathering rather than better decisions. Moderate costs give the best balance: $\eta=0.1$ reaches the best XD-Violence AP (72.43), while the default $\eta=0.5$ remains near-optimal on XD-Violence (72.29) and gives the best UCF-Crime AUC (84.46). In contrast, larger costs make the policy too conservative: at $\eta=2.0$, the interaction ratio decreases to 9.75%, but XD-Violence AP and UCF-Crime AUC fall to 69.90 and 82.53, respectively.

The $\lambda$ sweep is much flatter. Increasing $\lambda$ from 0.5 to 2.0 changes XD-Violence AP only within a 0.31-point range (71.98-72.29), while gradually increasing the interaction ratio, average steps, and distinct sampled locations. This indicates that once the cost term is present, the method is not highly sensitive to the exact AEI weight. The $N_{\mathrm{buf}}$ sweep shows a different pattern: a buffer of one observation is insufficient (70.29 AP), whereas $N_{\mathrm{buf}}=2$ and $N_{\mathrm{buf}}=3$ perform almost identically (72.29 vs. 72.33 AP). We therefore use $N_{\mathrm{buf}}=2$ as the default because it captures most of the benefit while avoiding unnecessary context growth.

*Table 13.* Trajectory construction variants on XD-Violence. AP is reported in %, Avg. Steps counts evidence-acquisition actions beyond the initial observation, and $\Delta$AP is measured relative to full iDPO.

| Variant | Ranking Signal | AP | Avg. Steps | $\Delta$AP |
|---|---|---|---|---|
| Terminal-only | Final prediction only | 66.87 | 0.2880 | -5.42 |
| Cost-agnostic | Task/evidence utility without interaction cost | 70.04 | 0.2865 | -2.25 |
| Extreme-pairs-only | Only highest- vs. lowest-utility pairs | 71.80 | 0.2690 | -0.49 |
| Full iDPO | Task + AEI + cost-aware trajectory ranking | **72.29** | 0.2650 | 0.00 |

## D.3. Trajectory Construction and Protocol Robustness

Through the ablations reported in Table 13, we next isolate the effect of preference construction for iDPO. Terminal-only ranking, which scores trajectories only by the final prediction, obtains 66.87 AP and is 5.42 points below full iDPO, showing that final-label correctness alone is too weak to supervise evidence acquisition and stopping. Removing the interaction cost improves over terminal-only ranking but still reaches only 70.04 AP, and it uses more steps than full iDPO (0.2865 vs. 0.2650), consistent with cost-agnostic preferences encouraging unnecessary exploration. Using only extreme utility pairs performs much better (71.80 AP), but remains 0.49 points below full iDPO, suggesting that the complete preference set provides useful additional comparisons. Overall, the dominant factor is whether the ranking signal captures task utility, AEI evidence quality, and interaction cost jointly.

*Table 14.* Stride robustness on XD-Violence. We use clip length $L$=16 and vary the evaluation stride $\Delta$ on the subsampled frame sequence. AP is reported in %.

| Stride $\Delta$ | Setting | XD AP |
|---|---|---|
| 1 | Dense overlap | 74.57 |
| 8 | Default | **72.29** |
| 16 | Non-overlapping | 69.89 |

When varying the evaluation stride for frame-level scoring (Table 14), dense overlap with $\Delta$=1 gives the highest AP (74.57), while non-overlapping evaluation with $\Delta$=16 is more challenging (69.89). The default $\Delta$=8 setting reaches 72.29 AP, providing a middle ground between dense temporal coverage and evaluation cost. Since prior methods usually report a single score under their own evaluation protocols and do not provide matched stride sweeps, we use the results in Table 1 only as reference. Under the default setting, Anom-$\pi$ remains higher than VERA at 70.11 AP and PANDA at 70.16 AP on XD-Violence. These results show that the advantage is not tied to a single stride choice, although denser evaluation naturally gives more opportunities to cover short anomalous intervals.

## D.4. Representative UCF-Crime Class-wise Evidence

Finally, the UCF-Crime class-wise breakdown (Table 15) highlights where active interaction helps most. The gain is much larger for Stealing (+13.31 AUC), where decisive cues are often local, subtle, and easy to miss under a fixed initial view. In contrast, RoadAccidents improves by a smaller +3.05 AUC because the abnormal event is usually global and visually salient, so the initial observation already provides stronger evidence. This helps explain why the dataset-level UCF-Crime improvement is moderate: active querying is most beneficial for localized or ambiguous categories, while gains are naturally diluted by classes that are already well served by open-loop observation.

*Table 15.* Representative class-wise gains on UCF-Crime. The examples illustrate where active querying helps more or less. Gains are reported as AUC(%) differences.

| Class | Typical Visual Pattern | AUC Gain |
|---|---|---|
| Stealing | Localized and subtle evidence | +13.31 |
| RoadAccidents | Global and visually salient events | +3.05 |

# E. Additional Qualitative Visualizations

We first provide paired comparison cases between Anom-$\pi$ and an open-loop QwenVL baseline, grouped into Fig. 6 and Fig. 7. We further include representative failure cases to illustrate remaining limitations of closed-loop evidence acquisition and stopping.

**Comparison cases I.**    The first pair of qualitative comparisons highlights cases where the abnormal cue is present but easy to under-localize (Fig. 6). In the street scene, the initial frames contain many distractors, including pedestrians, parked cars, and street lights. Anom-$\pi$ first probes the frames around the white car with a shattered windshield, then checks the previous segment to rule out a normal traffic scene and localizes the riot over the full clip. The open-loop baseline notices the broken window but treats it as a minor disturbance and predicts Normal. In the beach scene, the abnormality evolves from a tense approach to a struggle in shallow water. Anom-$\pi$ samples intermediate frames to connect the early confrontation with the later physical struggle, whereas the baseline fires only at the last few frames and misses the onset of the fight.

**Comparison cases II.**    The second pair shifts to cases that require either boundary refinement or close-up interaction evidence (Fig. 7). In the dim explosion sequence, early frames show a low-visibility military setting, but the decisive smoke and debris appear later. Anom-$\pi$ checks the suspected burst and its following context, extending the prediction to cover the visible aftermath; the baseline instead assigns the event to the beginning of the clip. In the residential scene, the global view looks like a routine encounter involving a uniformed person, so the initial hypothesis is Normal. Fine-grained sampling around the doorway reveals a person in a white shirt being physically restrained, which changes the decision to Fighting; the open-loop baseline misses this local contact and remains Normal.

**Failure cases.**    Fig. 8 illustrates two remaining limitations. In the redundant-exploration case, low visual quality and heavy blur make the additionally queried frames and previous-segment context nearly indistinguishable, so interaction increases cost without resolving the ambiguity. In the premature-stopping case, the agent inspects only a narrow local moment in a prison confrontation and does not further verify whether the tense interaction develops into violence. These cases suggest that future policies should better estimate information gain under poor visibility and calibrate uncertainty before triggering FINAL when ambiguity persists.

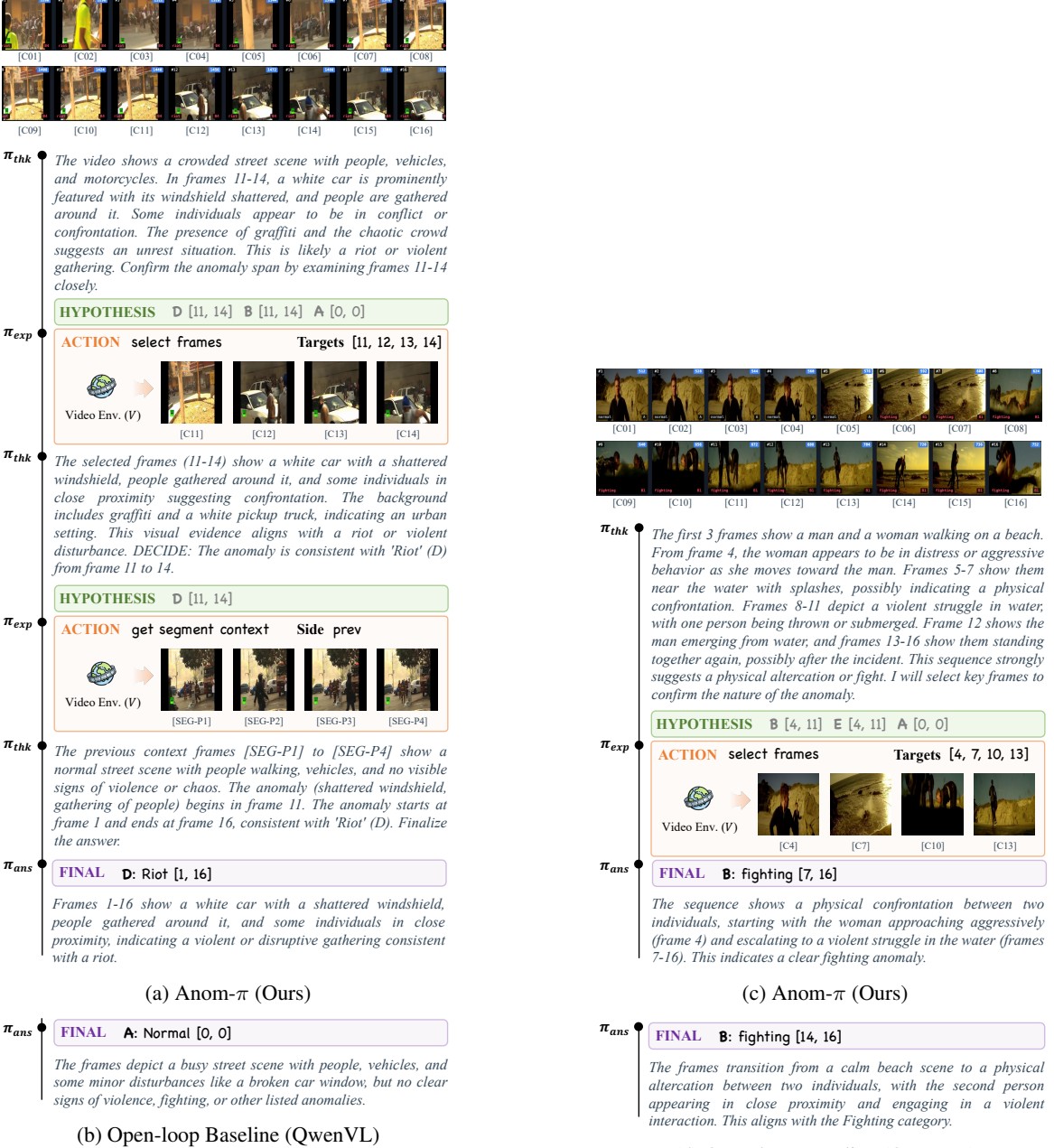

Figure 6. Qualitative comparison cases I. Anom-π is shown in (a,c), and the open-loop QwenVL baseline is shown in (b,d).

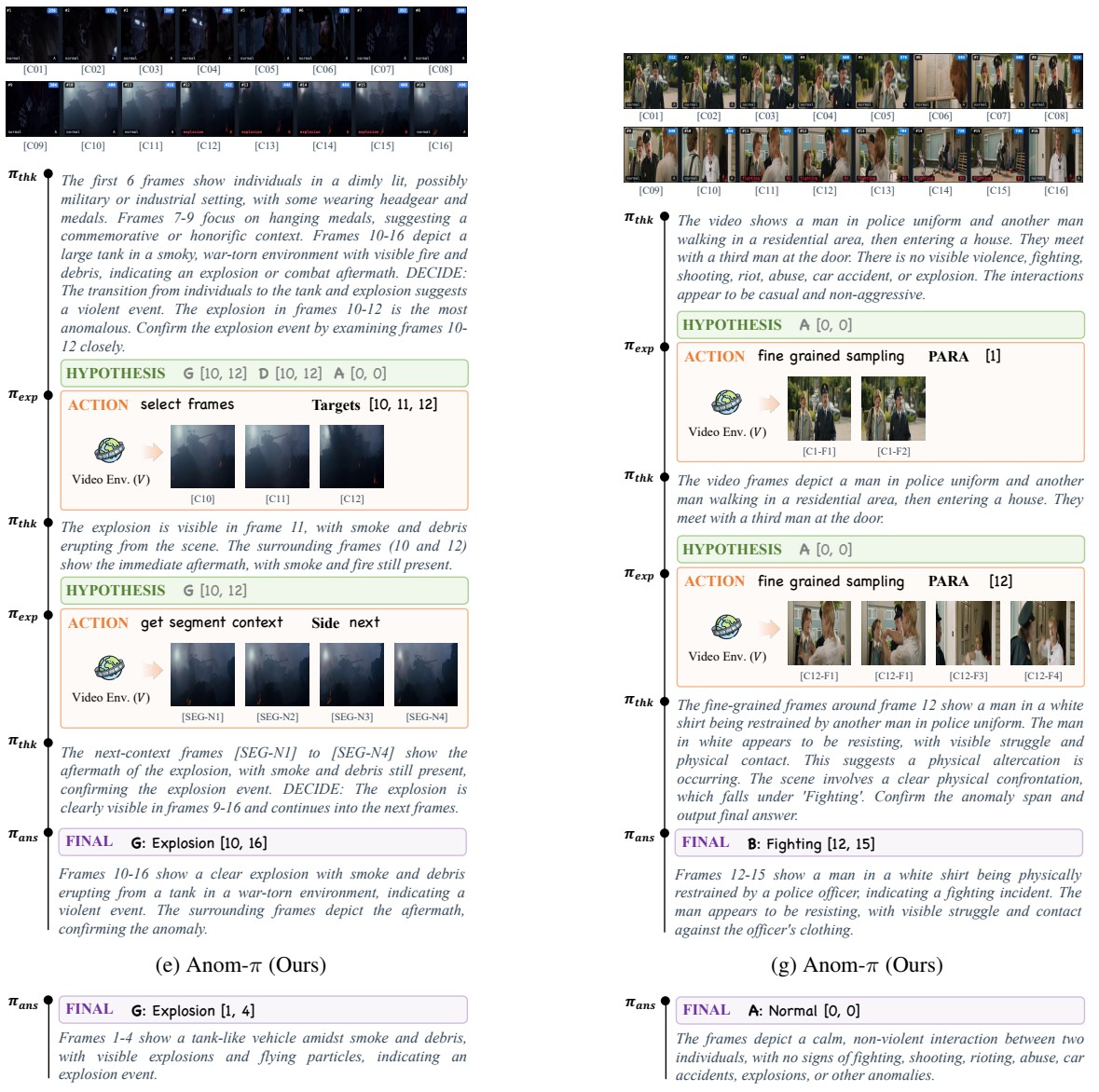

(e) Anom-$\pi$ (Ours)

(g) Anom-$\pi$ (Ours)

(f) Open-loop Baseline (QwenVL)

(h) Open-loop Baseline (QwenVL)

*Figure 7.* Qualitative comparison cases II. Anom-$\pi$ is shown in (e,g), and the open-loop QwenVL baseline is shown in (f,h).

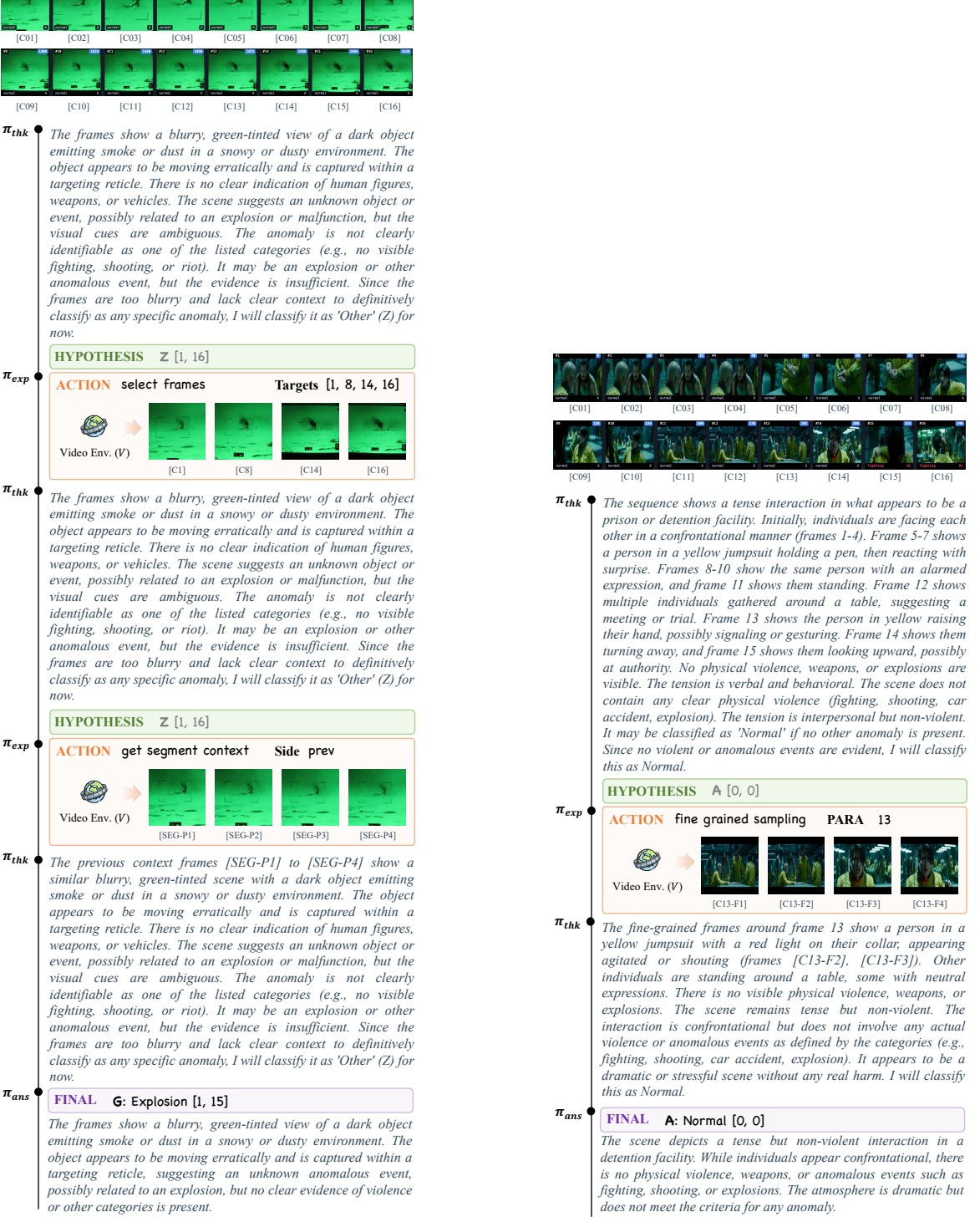

(a) Redundant exploration        (b) Premature stopping

*Figure 8.* Failure-case visualizations. (a) Redundant exploration under low visual quality. (b) Premature stopping under persistent temporal ambiguity.

