# OpenReview forum: "Learning to Watch: Active Video Anomaly Understanding via Interleaved Policy Optimization"
_ICML.cc/2026/Conference — ICML 2026 regular_

### Official Review · Reviewer_1Hxy · 2026-03-12

**Soundness:** 3
**Presentation:** 3
**Significance:** 3
**Originality:** 3
**Overall Recommendation:** 4
**Confidence:** 3

**Summary:**

The paper studies video anomaly understanding from an active, closed-loop perspective. Instead of reasoning over a fixed sampled video context, the proposed method, Anom-𝜋, learns an interaction policy over operators such as THINK, SAMPLE, EXPAND, BACKTRACK, and FINAL. Since fine-grained action supervision is unavailable, the paper constructs trajectory preferences using a handcrafted AEI utility and optimizes the policy using trajectory-level iDPO. Experiments on multiple anomaly understanding benchmarks suggest that active interaction can outperform passive inference, and the paper includes ablations and behavioral analyses to support the proposed design.

**Compliance With Llm Reviewing Policy:**

Affirmed.

**Final Justification:**

This rebuttal has addressed my main concerns.

**Key Questions For Authors:**

- How much of the observed gain comes from the learned interaction policy itself, versus the manually designed operator set and utility shaping?

- Can the authors provide a clearer controlled efficiency analysis at inference time (average interaction steps, queried frames/clips, and approximate latency/cost)?

- How sensitive is the method to the hand-designed utility coefficients and trajectory construction choices in AEI / iDPO?

- Can the authors clarify the frame-level scoring and aggregation protocol sufficiently to ensure fair comparison with prior work?

- Can the authors better characterize the settings in which active interaction helps most and the settings in which it helps less?

**Limitations:**

yes

**Strengths And Weaknesses:**

- Strengths: The paper has a clear and interesting central idea: anomaly understanding is naturally an evidence acquisition problem, not only a passive perception problem. This framing is compelling. I also appreciate that the paper includes a relatively broad empirical section with useful ablations and policy-behavior analyses, which makes the contribution feel more substantial than a superficial agent wrapper. The weakly supervised trajectory-level training approach is also pragmatic and well matched to the task.

- Weaknesses: The main weakness is the amount of manually designed structure in the system. The operator set, utility components, hypothesis machinery, novelty filtering, and interaction heuristics all play important roles, and the paper does not yet cleanly separate what is learned from what is engineered. I also view the theory section as suggestive rather than strongly justificatory. Finally, the evaluation protocol, especially the frame level scoring and aggregation procedure, needs clearer exposition to make the comparisons fully convincing.

---

> ### Author Rebuttal · Authors · 2026-03-31
>
> We thank the reviewer for recognizing the strength of our framing (`"This framing is compelling"`), our empirical ablations and the depth of our empirical study (`"more substantial than a superficial agent wrapper"`), and the practicality of our training design (`"pragmatic and well matched to the task"`).
> We have carefully addressed your questions below and incorporate these clarifications in the revision.
>
> > **Q1/W1: Learned Policy vs. Manual Design**
>
> (a) What is engineered vs. learned?
>
> We agree that the operator set, hypothesis tracker, novelty filtering, and AEI utility provide carefully designed inductive biases that define the interaction space and training signal. However, the core learning challenge remains substantial: under this fixed scaffold, the policy still has to learn which operator to use, where to query, when to revise, and when to stop.
>
> (b) How much gain comes from policy learning?
>
> A ZERO-SHOT baseline with the same toolbox but no interaction training achieves 34.92% AP on XD-Violence (Table 4), while iDPO reaches 72.29% AP. This suggests that most of the gain comes from learning to use the scaffold rather than from the scaffold alone. Tool removals in Table 5 also support this, as the removal of THINK causes a 6.75% drop.
>
> > **Q2: Inference Efficiency Analysis**
>
> (a) Interaction overhead across datasets
>
> |Dataset|Avg. Steps|Extra Frames|Latency / Clip (s)|
> |-|-|-|-|
> |XD|0.265|1.915|2.503|
> |UCF|0.246|1.489|2.077|
> |UBnormal|0.180|1.011|1.492|
> |CSAD|0.397|3.826|3.877|
>
> Average interaction is only 0.18-0.40 steps and 1.0-3.8 extra frames per clip, increasing from UBnormal to CSAD, consistent with our design that interaction is invoked more when clips are harder and more ambiguous.
>
> (b) Latency comparison on XD-Violence
>
> |Metric|LAVAD|VERA|Anom-π|
> |-|-|-|-|
> |Avg. Time / Clip (s)|3.912|2.882|2.503|
>
> Under the same H800 setting, Anom-π is faster than the larger VERA at the clip level (2.503s vs. 2.882s), partly because it uses a more compact 2B backbone and queries evidence only on demand.
>
> > **Q3: Sensitivity to Utility Coefficients and Trajectory Construction**
>
> (a) Utility coefficients (η and λ)
>
> |η|Inter. (%)|Avg. Steps|Dist. Locs|HRR (%)|AP (%)|
> |-|-|-|-|-|-|
> |0.0|36.11|0.2865|1.8696|2.81|70.04|
> |0.1|13.26|0.2658|1.8631|2.61|72.43|
> |0.5|11.74|0.2650|1.8570|2.49|72.29|
> |1.0|11.26|0.2593|1.7809|1.78|70.87|
> |2.0|9.748|0.2439|1.5764|1.53|69.90|
>
> |λ|Inter. (%)|Avg. Steps|Dist. Locs|HRR (%)|AP (%)|
> |-|-|-|-|-|-|
> |0.5|10.66|0.2507|1.6820|2.37|71.98|
> |1.0|11.74|0.2650|1.8570|2.49|72.29|
> |2.0|12.59|0.2761|1.9913|2.68|72.15|
>
> The method is fairly stable around the default setting because these hyperparameters mainly regulate interaction frequency and cost trade-offs rather than the core anomaly scoring itself. Consistently, η=0.1-0.5 changes AP by only 0.14, and λ changes it by only 0.31 even across a 4x range.
>
> (b) Trajectory construction
>
> |Trajectory Construction Variant|Ranking Signal U(τ)|AP (%)|Avg. Steps|
> |-|-|-|-|
> |Default (Ours)|Full R_task + λR_aei - ηC|72.29|0.2650|
> |Terminal-only|R_task only|66.87|0.2880|
> |Cost-agnostic|R_task + λR_aei|70.04|0.2865|
> |Extreme pairs only|Full utility|71.80|0.2690|
>
> The main sensitivity comes from whether the ranking signal captures interaction quality. Terminal-only ranking drops AP from 72.29 to 66.87, whereas extreme-pairs-only gives 71.80, so utility design matters more than pair selection because it directly shapes query quality and stopping behavior.
>
> > **Q4/W3: Frame-Level Scoring and Evaluation Protocol**
>
> We apologize for any confusion regarding Section 5.1 and clarify that the final frame-level anomaly score is the mean over overlapping windows. We use sliding-window evaluation with length 16 and stride 8, matching prior work. Normal clips assign 0 to all frames. For anomalous clips, frames inside the annotated anomalous span receive 1.0, and other frames in the same clip receive 0.5.
>
> To ensure a fair comparison, we follow the official evaluation protocols for all baselines (e.g., VERA with stride 1 and LAVAD with stride 16). On XD-Violence, Anom-π attains 74.57 / 72.29 / 69.89 AP at strides 1 / 8 / 16, indicating that the performance gains remain robust across different stride settings.
>
> > **Q5: Where Active Interaction Helps Most/Less**
>
> Active interaction is most helpful in scenes with local, subtle, or transient anomalies, where a fixed initial view is ambiguous. For example, on UCF-Crime, performance on "Stealing" improves by +13.31%. The gain is smaller for global and visually prominent anomalies, where the initial observation is often already sufficient. For example, the gain on "RoadAccidents" is +3.05%. Anom-π bypasses interaction in 88.26% of XD-Violence clips, preserving efficiency on easier cases. Gains are also smaller when additional queries cannot recover much new evidence.

---

> > ### Author Rebuttal · Reviewer_1Hxy · 2026-04-03
> >
> > Thank you for your candid review. Your comments have addressed and clarified most of the questions and concerns I had during the review process. Therefore, I have decided to maintain my original score.

---

> > > ### Author Response · Authors · 2026-04-03
> > >
> > > Thank you for your careful review and for your thoughtful evaluation. We are glad that our rebuttal helped clarify your questions and concerns, and we appreciate your recognition of the motivation and the pragmatic design of the proposed active anomaly understanding framework. Your feedback has been very helpful, and we will incorporate these clarifications in the final version.

---

### Official Review · Reviewer_cK2v · 2026-03-13

**Soundness:** 3
**Presentation:** 3
**Significance:** 3
**Originality:** 2
**Overall Recommendation:** 4
**Confidence:** 3

**Summary:**

This paper addresses observation ambiguity and insufficient evidence acquisition in passive video anomaly understanding by proposing Anom-$\pi$, which formulates VAU as an active evidence-seeking process and further optimizes the policy under weak supervision with iDPO. The overall method is fairly complete, including the interaction mechanism, training objective, and a certain level of theoretical analysis. Experimental results show competitive performance across multiple VAU benchmarks, especially in complex scenarios and open-set settings. Overall, the paper offers a meaningful and insightful perspective on VAU from the viewpoint of active evidence acquisition.

**Compliance With Llm Reviewing Policy:**

Affirmed.

**Key Questions For Authors:**

1. **On the sensitivity of AEI hyperparameters:** How robust is Anom-$\pi$ to key parameters such as $\lambda$, $\eta$, and $N_{\text{buf}}$? Could the authors provide a more systematic sensitivity analysis and offer some guidance on tuning these parameters in new scenarios?

2. **On the realizability of the “effective policy” assumption:** The theoretical analysis relies on the policy being sufficiently effective. Could the authors further formalize this property? During training, are there any practical constraints or mechanisms that help encourage or enforce it?

3. **On domain generalization and real-world deployment:** The gains appear to be more limited on some datasets. Have the authors analyzed whether this is related to insufficient scene coverage during training or the quality of the preference signals? Looking ahead to deployment in real-world surveillance settings, how do the authors plan to improve robustness under distribution shift?

**Limitations:**

The authors provide a fairly thorough discussion of the study’s limitations and potential societal impacts, and I appreciate the inclusion of mitigation measures such as human-in-the-loop verification and privacy-aware data handling. However, the paper does not provide a detailed discussion of the sensitivity of AEI hyperparameters or the scope and applicability of the theoretical assumptions.

**Strengths And Weaknesses:**

**Strengths**

Meaningful problem setting: The paper moves anomaly understanding from passive observation toward active evidence selection, which is an interesting and potentially impactful direction.

Fairly complete framework: The interaction design, training objective, and empirical validation are reasonably well connected.

Solid experimental support: Results on multiple datasets and ablation studies generally support the main claims.

Reasonably clear novelty: The combination of active perception and preference optimization in VAU gives the work a recognizable level of originality.

**Weaknesses**

Here is the English translation:

1. **Generalization to real-world scenarios still needs further validation:** The current experiments are mainly conducted on public benchmarks, and the paper does not yet sufficiently demonstrate the model’s applicability to unconstrained real-world surveillance videos, such as those involving challenging viewpoints, severe occlusion, low-light conditions, or camera shake.

2. **The analysis of key hyperparameters is insufficient:** Parameters in AEI, such as $\lambda$, $\eta$, and $N_{\text{buf}}$, may have an important impact on performance, but the paper lacks a systematic sensitivity analysis. As a result, it remains unclear how difficult the method is to tune and how stable it is when transferred across datasets or scenarios.

3. **There is still a gap between the theoretical assumptions and practical training:** The discussion of the “effective policy” assumption is insightful, but its validity is currently supported more by empirical evidence than by formal characterization. A clearer explanation of how this property can be encouraged or ensured during training would make the theoretical part more convincing.

4. **The analysis of interpretability could be strengthened:** For an anomaly understanding task, it is important not only to improve final classification performance, but also to show whether the model can provide more accurate explanations that are better grounded in visual evidence. At present, the qualitative comparison in this aspect is still limited, and the advantages over open-loop baselines have not yet been clearly demonstrated.

5. **The novelty claims could be articulated more clearly:** Although the overall framework is quite novel, the paper could do a better job of clarifying the key incremental contribution of iDPO over standard DPO, as well as why the proposed atomic operations are particularly suited to this VAU setting compared with more general active visual strategies.

---

> ### Author Rebuttal · Authors · 2026-03-31
>
> We thank the reviewer for recognizing the significance of our problem setting (`"an interesting and potentially impactful direction"`), the completeness of our framework (`"fairly complete framework"`), and the strength of our evaluation (`"solid experimental support"`). Below, we address these questions and incorporate the corresponding clarifications in the revision.
>
> > **Q1/W2/L1: Hyperparameter Sensitivity**
>
> |η|Inter. (%)|Steps|Locs|HRR (%)|AP (%)|
> |-|-|-|-|-|-|
> |0.0|36.11|0.2865|1.8696|2.81|70.04|
> |0.1|13.26|0.2658|1.8631|2.61|72.43|
> |0.5|11.74|0.2650|1.8570|2.49|72.29|
> |1.0|11.26|0.2593|1.7809|1.78|70.87|
> |2.0|9.748|0.2439|1.5764|1.53|69.90|
>
> |λ|Inter. (%)|Steps|Locs|HRR (%)|AP (%)|
> |-|-|-|-|-|-|
> |0.5|10.66|0.2507|1.6820|2.37|71.98|
> |1.0|11.74|0.2650|1.8570|2.49|72.29|
> |2.0|12.59|0.2761|1.9913|2.68|72.15|
>
> |N_buf|Inter. (%)|Steps|Locs|HRR (%)|AP (%)|
> |-|-|-|-|-|-|
> |1|6.84|0.1819|1.0809|2.48|70.29|
> |2|11.74|0.2650|1.8570|2.49|72.29|
> |3|12.16|0.2683|1.8939|2.65|72.33|
>
> Overall, the results indicate a stable region around the default setting. λ causes only minor AP variation, N_buf stabilizes once set to 2 or 3, and η is the main factor controlling the interaction-performance trade-off.
>
> > **Q2/W3/L2: Effective-policy Assumption**
>
> To make Assumption 4.2 more precise in the revision, we state a sufficient condition that matches its expectation form. Let $Δ_t(h_{t-1},a)=D_{KL}(p_θ(\cdot|a,h_{t-1})\|p_{θ'}(\cdot|a,h_{t-1}))$. If, for each round $t$ and for $P_θ^π$-almost every history $h_{t-1}$, there exists an informative set $A_{info}(h_{t-1})$ such that $Δ_t(h_{t-1},a)≥δ$ for all $a∈A_{info}(h_{t-1})$ and $π(A_{info}(h_{t-1})|h_{t-1})≥ρ$, then $\mathbb{E} _ {h _ {t-1}\sim P_θ^π,\;a_t\sim π(\cdot|h_{t-1})}[Δ_t(h_{t-1},a_t)]≥ρδ>0$, so Assumption 4.2 holds with $c=ρδ$.
>
> AEI is not intended to enforce this condition explicitly; instead, it encourages it through empirical surrogates: $r_{inf}$ encourages non-redundant queries, $r_{hyp}$ rewards hypothesis refinement, and the cost term discourages redundant probing. iDPO then prefers trajectories with higher AEI and penalizes weak evidence acquisition or poor stopping. Thus, the training objective serves as an empirical proxy aligned with the assumption, rather than a formal guarantee.
>
> > **Q3/W1: Generalization and Robustness**
>
> Gains are larger on localized or ambiguous events and smaller on globally obvious ones, since interaction helps most when the initial view is insufficient. UCF-Crime fits this pattern: fewer training videos, longer temporal spans, and more diverse categories likely yield sparser scene coverage and noisier preference pairs, while salient anomalies leave less room for interaction.
>
> |Class|Baseline|Anom-π|Gain|
> |-|-|-|-|
> |RoadAccidents|83.98|87.03|+3.05|
> |Stealing|64.83|78.14|+13.31|
>
> This is reflected by the larger gain on Stealing (+13.31) than on RoadAccidents (+3.05). While UBnormal is not itself a deployment-scale benchmark, its 78.75 AUC still suggests robustness beyond the training categories, and deployment robustness could be further strengthened with perturbation-based preference construction and broader scene coverage.
>
> > **W4: Interpretability and Visual Grounding**
>
> Open-loop baselines explain from a fixed observation, so missing cues can make explanations generic or weakly grounded, whereas Anom-π explains after explicit evidence acquisition, making outputs more traceable to queried frames. This is illustrated by Fig. 3's explosion example, where the initial view shows only a blurry flash near a stationary car, but rather than finalizing, Anom-π queries around frame 14, reveals smoke and debris, and updates the explanation to an explosion.
> Additional examples are available at https://anonymous.4open.science/r/anom-pi/Vis_for_Reviewer_cK2v.png.
>
> > **W5: Clarification on Novelty**
>
> (a) iDPO vs. a vanilla DPO baseline
>
> The key novelty of iDPO is to augment vanilla DPO with the AEI utility, so that trajectory preferences explicitly supervise evidence acquisition, hypothesis refinement, and stopping behavior in interleaved VAU. Vanilla DPO already improves over SFT through pairwise trajectory comparison, while iDPO further improves AP by 5.72 and reduces premature finalization by 0.081 by shaping the interaction process itself.
>
> |Method|Training signal|Premature|AP (%)|
> |-|-|-|-|
> |SFT|Single trajectory|0.412|56.86|
> |DPO (w/o curiosity)|Trajectory preference|0.358|66.57|
> |iDPO (ours)|Trajectory preference + AEI|0.277|72.29|
>
> (b) Why these operators are suited to VAU
>
> Our operators are tailored to the uncertainty structure of VAU, where the main challenge is temporal disambiguation of short-lived and context-dependent evidence rather than generic spatial search. Accordingly, Backtrack, Expand, and Sample support revisiting, contextualizing, and verifying suspicious moments, while Think/Final support revision and calibrated stopping; the ablations are consistent with this design, with the largest drop coming from removing Sample.

---

### Official Review · Reviewer_SpQp · 2026-03-13

**Soundness:** 4
**Presentation:** 4
**Significance:** 3
**Originality:** 4
**Overall Recommendation:** 5
**Confidence:** 4

**Summary:**

The paper focuses on the field of video anomaly understanding by shifting from a passive inference paradigm to an active, closed-loop hypothesis-verification process. It proposes Anom-$\pi$, a framework that interleaves cognitive reasoning with strategic evidence acquisition via atomic temporal operators like backtracking and fine-grained sampling. The authors address the challenge of learning such complex interaction strategies under video-level weak supervision by introducing interactive direct preference optimization. Extensive experiments were conducted on four benchmarks: XD-Violence, UCF-Crime, UBnormal, and CSAD. The results demonstrate that the proposed 2B-parameter model significantly outperforms much larger state-of-the-art models by efficiently gathering informative evidence.

**Compliance With Llm Reviewing Policy:**

Affirmed.

**Key Questions For Authors:**

1. Could you provide a detailed breakdown of the average inference time per clip compared to one-shot baselines like VERA, and discuss potential optimizations for real-time deployment?
2. How does the performance vary when the interaction cost $\eta$ is significantly increased or decreased, and is there a *sweet spot* that generalizes across different datasets?
3. Beyond the quantitative results, what are the most common types of premature stopping or redundant exploration behaviors still observed in the iDPO-trained agent?

**Limitations:**

yes

**Strengths And Weaknesses:**

**Strengths**
1. The paper introduces a novel active exploration paradigm that mimics human video-reviewing behavior, effectively addressing the observational aliasing problem inherent in static sampling.
2. The proposed iDPO algorithm provides a robust solution for training complex interactive policies using only video-level weak supervision, which is highly practical for real-world scenarios.
3. The model achieves state-of-the-art performance across multiple benchmarks with remarkable parameter efficiency, outperforming 72B-scale models with only 2B parameters.

**Weaknesses**
1. The paper does not fully explore the sensitivity of the performance to the interaction cost hyperparameters, which could significantly affect the balance between accuracy and latency.
2. The iterative nature of the interleaved policy may lead to increased inference latency compared to one-shot models, which might limit its applicability in strictly real-time monitoring systems.
3. The improvement on the UCF-Crime dataset is relatively modest compared to other benchmarks, suggesting the policy might struggle with certain domain-specific scene patterns.

---

> ### Author Rebuttal · Authors · 2026-03-31
>
> We sincerely thank the reviewer for recognizing the novelty of our work (`"novel active exploration paradigm"`), the practicality of our iDPO design (`"highly practical for real-world scenarios"`), and the efficiency of Anom-π (`"remarkable parameter efficiency"`). Detailed responses to your questions are provided below.
>
> > **Q1/W2: Inference Efficiency**
>
> (a) Average Latency per Clip
>
> ||LAVAD|VERA|Anom-π|
> |-|-|-|-|
> |Avg. Time / Clip (s)|3.912|2.882|2.503|
>
> Overall, Anom-π averages 2.503 s/clip, compared with 2.882 s/clip for VERA and 3.912 s/clip for LAVAD. The key reason is that interaction is invoked only on ambiguous clips, while easy clips are handled directly without extra evidence acquisition. As a result, the amortized latency overhead remains limited in practice.
>
> (b) Step-level Latency Breakdown of Anom-π
>
> ||THINK|BACKTRACK|EXPAND|SAMPLE|FINAL|
> |-|-|-|-|-|-|
> |Avg. Time / Action (s)|1.5097|1.7473|1.7636|1.7353|0.8638|
> |Avg. Cumulative Time / Sample (s)|1.1779|0.0478|0.0665|0.3470|0.8638|
>
> The table above explains why the overall latency remains low. Although each interactive action is not negligible in isolation, only 11.74% of clips trigger interaction, so the cumulative overhead from BACKTRACK/EXPAND/SAMPLE is modest after amortization.
>
> (c) Optimizations for Real-Time Deployment
>
> Our current implementation already uses a KV cache, so once the initial frames and text history are encoded, each subsequent step only introduces a small number of new tokens.
>
> While the current system is not yet optimized for hard real-time deployment, the framework is structurally compatible with several standard engineering optimizations:
> - Parallel Batch Inference: Treating sliding-window clips as independent episodes enables seamless batching across multiple streams to maximize GPU throughput.
> - Rolling KV-Cache: Given our clip length of 16 and stride of 8, inheriting visual tokens for the 8 overlapping frames from the previous window eliminates redundant encoding.
> - Low-Bit Quantization: With our compact 2B-parameter backbone, 4/8-bit quantization could further reduce memory bandwidth during deployment.
>
> > **W1/Q2: Hyperparameter Sensitivity**
>
> |η|Inter. Ratio(%)|Avg. Steps|Dist. Locs|XD/AP(%)|UCF/AUC(%)|
> |-|-|-|-|-|-|
> |0.0|36.11|0.2865|1.8696|70.04|84.52|
> |0.1|13.26|0.2658|1.8631|72.43|83.22|
> |0.5|11.74|0.2650|1.8570|72.29|84.46|
> |1.0|11.26|0.2593|1.7809|70.87|82.87|
> |2.0|9.748|0.2439|1.5764|69.90|82.53|
>
> As η decreases, the agent explores more aggressively, with a higher interaction ratio and more sampled locations; as η increases, it becomes more conservative, and performance drops due to earlier stopping. The results suggest a fairly broad stable region: moderate values (η=0.1-0.5) consistently provide the best cross-dataset trade-off, while more extreme settings are less effective. We choose η=0.5 because it achieves the best UCF-Crime result (84.46) while remaining near-optimal on XD-Violence (72.29).
>
> > **Q3: Qualitative Analysis of Agent Behaviors**
>
> (a) Typical Redundant Exploration: Redundant exploration is most common in deceptively hard clips with extremely weak anomalous cues, such as small distant actions or subtle shoplifting signals. In these cases, each additional query provides limited new evidence, so the agent may repeatedly invoke BACKTRACK or SAMPLE before its confidence crosses the stopping threshold.
>
> (b) Typical Premature Stopping: Premature stopping tends to occur in highly dynamic but ambiguous moments, where salient local motion creates an early but incomplete hypothesis. In such cases, the agent may terminate before EXPAND reveals the broader temporal context, indicating occasional overconfidence in local cues.
>
> The visual examples are available at https://anonymous.4open.science/r/anom-pi/Vis_for_Reviewer_SpQp.png.
>
> > **W3: Analysis of Performance Gains on UCF-Crime**
>
> (a) Dilution from Salient Anomalies
>
> UCF-Crime contains many global anomalies (e.g., road accidents) where a single open-loop static view already achieves high confidence. Our active policy significantly improves more localized or subtle anomalies (e.g., stealing) that suffer from observation aliasing, but this gain is diluted in the dataset-wide average.
>
> |Classes|Baseline|Anom-π|Gain|
> |-|-|-|-|
> |RoadAccidents|83.98|87.03|+3.05|
> |Stealing|64.83|78.14|+13.31|
>
> (b) Perception Bottleneck on Extreme Cases
>
> UCF-Crime also includes domain-specific scenarios with extremely weak features (e.g., tiny motion, heavy occlusion) over long durations. In such cases, active querying can still help, but further gains likely also require stronger fine-grained visual perception from the backbone. This suggests that the remaining gap reflects both policy and perception limits when the visual evidence itself is extremely weak.

---

> > ### Author Rebuttal · Reviewer_SpQp · 2026-04-04
> >
> > My concerns have been adequately addressed and I have decided to maintain my original score.

---

> > > ### Author Response · Authors · 2026-04-04
> > >
> > > We sincerely appreciate your positive feedback and strong support for our work. We are glad that our rebuttal adequately addressed your questions and concerns, and your recognition of the motivation and effectiveness of the proposed paradigm is very encouraging to us. Thank you again for your time and professional review.

---

### Official Review · Reviewer_x7C1 · 2026-03-13

**Soundness:** 2
**Presentation:** 3
**Significance:** 3
**Originality:** 3
**Overall Recommendation:** 4
**Confidence:** 4

**Summary:**

The proposed Anom-π framework reformulates video anomaly understanding as an active, closed-loop sequential decision-making process. It simulates human video inspection by alternating cognitive reasoning (THINK) with atomic interaction operations (INTERACT)—such as temporal backtracking, expansion, and fine-grained sampling—to overcome the observation ambiguity caused by traditional passive sampling.

To train this policy under weak supervision with only video-level labels, the authors introduce Interactive Direct Preference Optimization (iDPO). Using an Active Evidence Inquiry (AEI) utility function, the method aligns preferences across different exploration trajectories, encouraging the model to terminate reasoning rationally once sufficient evidence is obtained.

Experiments show that with only 2B parameters and minimal context (16–32 frames), the model significantly outperforms larger state-of-the-art models on benchmarks such as XD-Violence and CSAD, achieving efficient and interpretable anomaly localization.

**Compliance With Llm Reviewing Policy:**

Affirmed.

**Final Justification:**

The experiments in rebuttal help to validate the effectiveness of this paper, so I raise my rating.

**Key Questions For Authors:**

Please answer the three questions in the weakness part. If the questions are reasonably answered I will raise my rating.

**Limitations:**

yes

**Strengths And Weaknesses:**

Strength:
1. It's promising to mimic human behavior to actively explore a video with the innovative active inference paradigm
2. The AEI training function skillfully guiding the model to perform rational termination once sufficient evidence is gathered, preventing redundant exploration.
3. Despite its compact model scale, Anom-pi demonstrates strong competitiveness and robustness across multiple benchmarks.

Weaknesses
1. Lack of Detailed Behavioral Statistics and Trajectory Analysis
The manuscript claims that Anom-pi mimics human-like video reviewing behavior through strategic evidence acquisition. However, the current evaluation lacks a granular statistical breakdown of the agent's decision-making process across different datasets.
Specifically:
- Action Distribution: There is no reported data on the individual usage rates for the defined actions—BACKTRACK, EXPAND, and SAMPLE—across different benchmarks.
- Interaction Frequency: The average number of actions invoked per video segment is not provided, making it difficult to assess the actual computational overhead.
- Information Gain and Exploration: Regarding the Curiosity Signal, the authors should provide the average number of distinct temporal locations sampled per trajectory.
- Hypothesis Evolution Reliability: While the paper mentions belief refinement ($r_{hyp}$), it is unclear how often a model actually revises a hypothesis and subsequently seeks new evidence to verify that change.
- Rationality vs. Reward Hacking: Without evidence showing that trajectory length correlates with case difficulty, there is a risk that the agent is simply exploring for a fixed number of steps to maximize rewards rather than practicing rational termination based on the Value of Information.

2. Limited Model Scale and Generalization Concerns
The authors exclusively utilize Qwen3-VL-Instruct-2B as the backbone for all experiments. While the 2B model achieves competitive results, using only a small-scale model makes it difficult to determine if the proposed iDPO framework and active exploration paradigm scale effectively to larger, more powerful vision-language models. Testing on a single, relatively small backbone limits the proof of the framework’s robustness and its ability to generalize across different architectures or parameter scales in complex, open-world scenarios.

3. Absence of Comparison with Advanced Optimization Algorithms
The paper introduces Interactive Direct Preference Optimization (iDPO) to align the policy. The authors do not provide a comparison with Group Relative Policy Optimization (GRPO). It remains unclear why iDPO was selected over GRPO and how it compares in terms of training stability, sample efficiency, or the quality of the final interleaved policy.

---

> ### Author Rebuttal · Authors · 2026-03-31
>
> We thank the reviewer for recognizing the promise of our work (`"promising to mimic human behavior"`), the effectiveness of our AEI design (`"skillfully guiding the model to perform rational termination"`), and the competitiveness of Anom-π (`"strong competitiveness and robustness across multiple benchmarks"`). We have carefully considered your concerns and provide detailed responses below.
>
> > **W1: Behavioral Statistics & Trajectory Analysis**
>
> (a) Action Distribution, Interaction Frequency, and Information Gain
>
> |Dataset|BACKTRACK(%)|EXPAND(%)|SAMPLE(%)|Avg. steps|Distinct locs|Extra frames|Latency (s)|
> |-|-|-|-|-|-|-|-|
> |XD|10.33|14.22|75.46|0.265|1.857|1.915|2.503|
> |UCF|41.71|12.44|45.85|0.246|1.334|1.489|2.077|
> |UB|25.92|21.14|52.94|0.180|0.852|1.011|1.492|
> |CSAD|15.77|20.49|63.73|0.397|3.426|3.826|3.877|
>
> The action mix varies across benchmarks. SAMPLE dominates on XD, UB, and CSAD (75.46%, 52.94%, and 63.73%), especially on XD where rapid scene transitions make fine-grained sampling useful for transient evidence. UCF shows much more BACKTRACK (41.71%), consistent with its long continuous surveillance videos, while overall interaction overhead remains modest, with fewer than 0.40 interaction steps and at most 3.88 s latency per clip.
>
> (b) Hypothesis Evolution Reliability
>
> |Dataset|Interactive Ratio (%)|Revision Rate (% overall; within Interactive)|
> |-|-|-|
> |XD|11.74|2.49 (21.20)|
> |UCF|16.52|3.32 (20.10)|
> |UB|12.10|0.69 (5.72)|
> |CSAD|37.43|11.72 (31.31)|
>
> Revision followed by further evidence seeking occurs in 21.20% of interactive trajectories on XD-Violence, 20.10% on UCF-Crime, and 31.31% on CSAD, but only 5.72% on UBnormal. Since the interaction ratio remains low on XD/UCF/UB (11.74%-16.52%), the model usually stops directly on easy cases and revises only when more evidence is needed.
>
> (c) Rationality vs. Reward Hacking
>
> |Dataset|Avg. steps (overall)|Avg. steps (challenge)|Avg. extra frames (overall)|Avg. extra frames (challenge)|
> |-|-|-|-|-|
> |XD|0.27|0.56|1.92|2.37|
> |UCF|0.25|0.42|1.49|2.12|
>
> Following PANDA, we also report statistics on its manually defined challenge subset. Compared with the overall average, challenge cases require more interaction on both datasets, with average steps increasing from 0.27 to 0.56 on XD-Violence and from 0.25 to 0.42 on UCF-Crime. This suggests that the agent allocates more interaction to harder cases rather than following a fixed exploration budget.
>
> > **W2: Model Scalability and Generalization**
>
> We further test Anom-π with two additional backbones on XD-Violence: Qwen3-VL-8B-Instruct, a larger model from the same family, and InternVL3_5-2B-Instruct, a model from a different VLM family. In both cases, Anom-π improves over the corresponding base model, suggesting that the gain is not specific to a single model family or architecture.
>
> |Method|AP|Δ AP(%)|AUC|Δ AUC(%)|
> |-|-|-|-|-|
> |Qwen3-VL-8B-Instruct|58.83|-|82.35|-|
> |w/ Anom-π|74.46|15.63|93.47|11.12|
> |InternVL3_5-2B-Instruct|44.78|-|72.34|-|
> |w/ Anom-π|66.18|21.40|89.24|16.90|
>
> > **W3: Why Not GRPO?**
>
> GRPO is a strong baseline, and under weakly supervised VAU with only video-level labels and long interleaved trajectories, we believe iDPO is better matched to the available supervision for three reasons.
>
> (a) Task Fit
>
> - In VAU, two interleaved trajectories can reach the same final label while differing in evidence quality, redundancy, verification, and stopping. This makes pairwise trajectory preference, as used by iDPO, more natural than the scalar reward required by GRPO.
> - iDPO directly aligns chosen/rejected trajectories ranked by AEI utility, so it optimizes the behavior we care about in this task, namely well-justified evidence acquisition and stopping, rather than only the final label.
> - Under weak supervision with only video-level labels and multi-step rollouts, GRPO would still need to collapse these trajectory-level differences into scalar rewards and advantages, which can make optimization more sensitive to reward noise and difficult credit assignment.
>
> (b) Practical Considerations
>
> In our setting, GRPO would require converting trajectory-level scores into scalar advantages for long rollouts, which makes optimization more sensitive to reward noise and credit assignment. It is also substantially more expensive on the same H800 setup, requiring about 140 GB peak VRAM and over 34 hours per epoch, versus about 54 GB and 2 hours for iDPO. In contrast, iDPO is more directly aligned with the target interleaved policy because it explicitly prefers better query-revise-stop behavior.
>
> (c) Broader Discussion
>
> We appreciate the value of recent GRPO-based works such as VAD-R1, VAU-R1, A2Seek, IAD-R1, CueBench, and JUDO, as well as the substantial effort required to build richer reasoning supervision for these directions. Our method is complementary to these works, as we focus on a weaker but practical setting with only video-level labels. We will further expand this comparison and discussion in the revision.

---

> > ### Author Rebuttal · Reviewer_x7C1 · 2026-04-04
> >
> > Given the newly provided results, I would like to weak accept this paper.

---

> > > ### Author Response · Authors · 2026-04-04
> > >
> > > Thank you very much for raising your score and for your positive feedback. We sincerely appreciate your careful consideration of our rebuttal and are glad that it addressed your concerns. Your constructive feedback has been very helpful in strengthening the paper.

---

### Decision · Program_Chairs · 2026-04-30

**Decision:**

Accept (regular)

**Comment:**

This paper presents work on video anomaly detection.  The core contribution is a sequential decision making process that incorporates active reasoning and atomic interaction methods.  The reviewers appreciated the active analysis of video method, the solid empirical results, and the effectiveness with a relatively small model size.

Minor concerns over details in the methods and experimental setups were raised in the initial reviews.  These were largely addressed in the author responses.

Given the solid algorithmic contribution and advances in active understanding of video anomalies, this paper is recommended for acceptance.